# FP4DiT: Towards Effective Floating Point Quantization for Diffusion Transformers

**Ruichen Chen**[*]                                                                                            *ruichen1@ualberta.ca*
*Department of Electrical and Computer Engineering, Faculty of Engineering*
*University of Alberta*

**Keith G. Mills**[*]                                                                                            *keith.mills@lsu.edu*
*Division of Computer Science and Engineering, College of Engineering*
*Louisiana State University*

**Di Niu**                                                                                            *dniu@ualberta.ca*
*Department of Electrical and Computer Engineering, Faculty of Engineering*
*University of Alberta*

**Reviewed on OpenReview:** *https://openreview.net/forum?id=CcnH4mSQbP*

## Abstract

Diffusion Models (DM) have revolutionized the text-to-image visual generation process. However, the large computational cost and model footprint of DMs hinders practical deployment, especially on edge devices. Post-training quantization (PTQ) is a lightweight method to alleviate these burdens without the need for training or fine-tuning. While recent DM PTQ methods achieve W4A8 (i.e., 4-bit weights and 8-bit activations) on integer-based PTQ, two key limitations remain: First, while most existing DM PTQ methods evaluate on classical DMs like Stable Diffusion XL, 1.5 or earlier, which use convolutional U-Nets, newer Diffusion Transformer (DiT) models like the PixArt series, Hunyuan and others adopt fundamentally different transformer backbones to achieve superior image synthesis. Second, integer (INT) quantization is prevailing in DM PTQ but does not align well with the network weight and activation distribution, while Floating-Point Quantization (FPQ) is still under-investigated, yet it holds the potential to better align the weight and activation distributions in low-bit settings for DiT. In this paper, we introduce FP4DiT, a PTQ method that leverages FPQ to achieve W4A6 quantization. Specifically, we extend and generalize the Adaptive Rounding PTQ technique to adequately calibrate weight quantization for FPQ and demonstrate that DiT activations depend on input patch data, necessitating robust online activation quantization techniques. Experimental results demonstrate that FP4DiT achieves higher CLIP, ImageReward and HPSv2 performance compared to integer-based PTQ at the W4A6 and W4A8 precision levels while generating convicning visual content on PixArt-$\alpha$, PixArt-$\Sigma$ and Hunyuan. Code is available at https://github.com/cccrrrccc/FP4DiT.

## 1 Introduction

Diffusion Transformers (DiT) (Peebles & Xie, 2023) are on the forefront of open-source generative visual synthesis. In contrast to earlier text-to-image (T2I) Diffusion Models (DMs) like Stable Diffusion v1.5 (Rombach et al., 2022) and Stable Diffusion XL (Podell et al., 2024) that utilize a classical U-Net structure, DiTs such as PixArt-$\alpha$ (Chen et al., 2024b), PixArt-$\Sigma$ (Chen et al., 2024a) and Stable Diffusion 3 (SD3) (Esser et al., 2024) leverage streamlined, patch-based Transformer architectures to generate high-resolution images.

Nevertheless, similar to U-Nets, DiTs utilize a lengthy denoising process that incurs a high computational inference cost. Approaches to reducing this cost generally aim to reduce the inference cost of the denoising

---

[*]Equal contribution.

network (Fang et al., 2023; Castells et al., 2024) or reduce the number of denoising steps (Yin et al., 2024; Salimans & Ho, 2022). One specific method to reduce the denoiser cost is quantization (Dettmers et al., 2022; Yao et al., 2022), which reduces the bit-precision of neural network weights and activations. As the first Post-Training Quantization (PTQ) schemes for DMs, PTQ4DM (Shang et al., 2023b) and Q-Diffusion (Li et al., 2023) demonstrate that the range and distribution of U-Net activations crucially depend on the diffusion timestep. More recent state-of-the-art works like TFMQ-DM (Huang et al., 2024) specialize quantization for U-Net timestep conditioning which may not generalize to newer DiTs. Further, methods like ViDiT-Q (Zhao et al., 2024) adapt outlier suppression technique (Xiao et al., 2023) to DiTs, but overlook broader advantages of prior DM PTQ like weight reconstruction (Nagel et al., 2020; Li et al., 2021).

Moreover, the prevailing datatypes in existing DM PTQ literature (Shang et al., 2023b; Li et al., 2023; Huang et al., 2024; Zhao et al., 2024; So et al., 2024; Feng et al., 2025; Mills et al., 2025) are integer-based (INT), which provide uniformly distributed values (Nahshan et al., 2021) unlike the non-uniform distribution of weights and activations in modern neural networks (Shen et al., 2024). Thus, PTQ for text-to-image (T2I) DiTs below W4A8 precision (4-bit weights and 8-bit activations) without severely compromising generation quality remains an open challenge.

In this paper, we present FP4DiT, which achieves W4A6 PTQ on Diffusion Transformers with non-uniform Floating-Point Quantization (FPQ) (Kuzmin et al., 2022), thus achieving high quantitative and qualitative T2I performance. Besides, by introducing FPQ, FP4DiT not only aligns the quantization levels better with the weight and activation distribution with negligible computational overhead, it also massively reduces the cost of weight calibration by over $8\times$. Our detailed contributions are summarized as follows:

1. We apply FPQ to DiT to address the misalignment between the existing DM PTQ literature and the non-uniform distribution of network weights and activations.

2. We reveal the critical role of preserving the sensitive interval of DiT's GELU activation function and propose a mixed-format FPQ method tailored for DiT.

3. We examine the adaptive rounding (AdaRound) (Nagel et al., 2020) mechanism, originally designed for integer PTQ, and reveal a performance-hampering design limitation when applied to FPQ. In this paper, we introduce a novel mathematical scaling mechanism that greatly improves the performance of AdaRound when utilized in the FPQ scenario.

4. We analyze DiT activation distributions and visualize how they contrast to those of convolutional U-Nets, especially with respect to diffusion timesteps. Specifically, while U-Net activation ranges *shrink* with timestep progression, DiT activations ranges instead *shift* over time. To address this, we implement an effective online activation quantization (Wu et al., 2023b; Yao et al., 2022) scheme to accommodate DiT activations.

We apply FP4DiT as a PTQ method on T2I DiT models, namely PixArt-$\alpha$, PixArt-$\Sigma$, and Hunyuan. To verify the effectiveness of FP4DiT, we conduct experiments on the MS-COCO dataset (Lin et al., 2014b) and the Human Preference Score v2 (HPSv2) T2I benchmark, to outperform existing methods like Q-Diffusion (Li et al., 2023), TFMQ-DM (Huang et al., 2024), ViDiT-Q (Zhao et al., 2024) and Q-DiT (Chen et al., 2025a) at the W4A8 and W4A6 precision levels. Additionally, we perform a human preference study which demonstrates the superiority of FP4DiT-generated images.

## 2 Related Work

Diffusion Transformers (DiT) (Peebles & Xie, 2023) replace the classical convolutional U-Net (Rombach et al., 2022) backbone with a modified Vision Transformer (ViT) (Dosovitskiy et al., 2020) to increase scalability. Although the introduction of DiT architectures in newer DMs (Chen et al., 2024b;a; Esser et al., 2024; Li et al., 2024; Labs; Xie et al., 2024) enables the generation of high-quality visual content (Brooks et al., 2024), DiTs still suffer from a computationally expensive diffusion process, rendering deployment on edge devices impractical and cumbersome. Methods to reduce this cost might aim to compress the denoiser neural network by pruning (Fang et al., 2023; Castells et al., 2024), sparsifying key operations (Chen et al.,

2025b), or by targeting a reduction in the number of denoising steps needed to generate an image (Yin et al., 2024; Salimans & Ho, 2022). This work focuses on the former objective while also aiming to discover some of the challenges associated with achieving DiT improvement compared to prior U-Nets.

Quantization is a neural network compression technique that involves reducing the bit-precision of weights and activations to lower hardware metrics like model size, inference latency and memory consumption (Nagel et al., 2021). The objective of quantization research is to reduce bit-precision as much as possible while preserving overall model performance (Ma et al., 2024). There are two main classes of quantization: Quantization-Aware-Training (QAT) (Sui et al., 2025; He et al., 2023; Feng et al., 2025) and Post-Training Quantization (PTQ) (Li et al., 2021; Mills et al., 2025; Frantar & Alistarh, 2022). Specifically, PTQ is more lightweight and neither requires re-training nor substantial amounts of data. Rather, PTQ requires a small amount of data to calibrate quantization scales (Nagel et al., 2020), typically in a block-wise manner (Li et al., 2021). However, while most PTQ methods rely on uniformly-distributed integer (INT) quantization techniques (Krishnamoorthi, 2018; Jacob et al., 2018; Zhang et al., 2025), recent literature highlights the advantages of low-bit floating point quantization (FPQ) (Wu et al., 2023b; Liu et al., 2023) for Large Language Models (LLM). Therefore, in this paper we investigate the challenges in applying FPQ to DM PTQ.

The denoising process of DMs brings new challenges for PTQ compared with traditional computer vision neural networks. The earliest DM PTQ research (Shang et al., 2023b) reveals the significant activation range changes across different denoising timesteps. Q-Diffusion (Li et al., 2023) samples calibration data across different denoising timesteps to address this challenge. TDQ (So et al., 2024) calibrates an individual set of the quantization parameters across different time steps, offering a more fine-grained approach to managing temporal dependencies. TFMQ-DM (Huang et al., 2024) highlights the sensitivity of temporal features in U-Nets and introduce a calibration objective aimed at better preserving temporal characteristics. However, the above works are specific to U-Net architectures while DiT architectures feature distinct activation characteristics.

In contrast, LLM quantization (Dettmers et al., 2022; Frantar et al., 2022; Chee et al., 2023; Liu et al., 2024; Zhang et al., 2024) operates on generative AI transformers. The key challenge is that multi-billion parameter transformers tend to generate outlier hidden states that are difficult to effectively quantize while preserving end-to-end performance (Lee et al., 2024a; Shang et al., 2023a). This can be addressed by leveraging the learnable affine shift of layernorm operations to adjust transformer attention and feedforward weights (Xiao et al., 2023; Lin et al., 2024b;a). However, DiTs use Adaptive Layernorm (AdaLN) (Perez et al., 2018), which ties the affine shift to the timestep embedding, so these methods are less applicable. Additionally, LLM quantization typically features more lightweight calibration (Ashkboos et al., 2024) as parameter-heavy models make advanced PTQ (Nagel et al., 2020; Li et al., 2021) costly. However, DiTs typically have fewer parameters and benefit from diverse calibration sets to cover multiple timesteps. Thus, FP4DiT leverages prior work on PTQ calibration to quantize DiTs.

Finally, some early research exists on DiT PTQ (Chen et al., 2025a). HQ-DiT (Liu & Zhang, 2024) apply FPQ to class-conditional ImageNet (Deng et al., 2009) DiTs, but do not consider text-to-image (T2I) models like the PixArt (Chen et al., 2024b;a) series. ViDiT-Q (Zhao et al., 2024) utilizes fine-grained techniques including channel balancing, mixed-precision and LLM outlier suppression, it does not incorporate weight reconstruction (Nagel et al., 2020; Li et al., 2021) from prior DM PTQ works. In contrast, this work aggregates knowledge from existing DM PTQ methods and refines them for application on T2I DiT models.

## 3 Methodology

In this section, we present our PTQ solution for the T2I DiT model. First, we analyze the sensitivity of the DiT block in the PixArt and Hunyuan model and propose a mixed FP format for the FP4 weight quantization. Second, we propose a scale-aware AdaRound tailored for FP weight quantization. Lastly, we investigate and contrast U-Net and DiT activation distribution information.

## 3.1 Uniform vs. Non-Uniform Quantization

Quantization compresses neural network size by reducing the bit-precision of weights and activations, e.g. rounding from 32/16-bit datatypes into an $n$-bit quantized datatype, where $n \leq 8$ typically. For instance, we can perform uniform integer (INT) quantization on a tensor $\mathbf{X}$ to round it into a lower-bit representation $\mathbf{X}^{(\text{int})}$ as follows:

$$\mathbf{X}^{(\text{int})} = \text{clip}\left(\left\lfloor \frac{\mathbf{X}}{s} \right\rceil + z, x_{\min}, x_{\max}\right) \tag{1}$$

where $s$ is scale, $z$ is the zero point, and $\lfloor \cdot \rceil$ is operation rounding-to-nearest. INT quantization rounds the values to a range with $2^n$ points. Specifically, the range is always a uniform grid, whose size decreases by half each time $n$ decreases by 1.

In contrast, Floating-Point Quantization (FPQ) uses standard floating-point numbers as follows:

$$f = (-1)^{d_s} 2^{p-b} \left(1 + \frac{d_1}{2} + \frac{d_2}{2^2} + \cdots + \frac{d_m}{2^m}\right) \tag{2}$$

where $d_s \in \{0, 1\}$ is the sign bit and $b$ is the bias. $p = d_1 + d_2 * 2 + \cdots + d_e * 2^{e-1}$ represents the $e$-bit exponent part while $\left(1 + \frac{d_1}{2} + \frac{d_2}{2^2} + \cdots + \frac{d_m}{2^m}\right)$ represents the $m$-bit mantissa part. Note that $d_i \in \{0, 1\}$ for bits in both the mantissa and the exponent part. The FP format can be seen as multiple consecutive $m$-bit uniform grids with different exponential scales. Therefore, the FPQ is operated similarly to Equation 1, with distinct scaling factors applied to values across varying magnitudes.

The key advantage of FPQ, especially at low-bit precision for quantization, is that they enjoy a richer granularity of value distributions owing to the numerous ways we can vary the allocation of exponent and mantissa bits. This is analogous to the introduction of the 'BFloat16' (Kalamkar et al., 2019) format, which achieves superiority over the older IEEE standard 754 'Float16' (Kahan, 1996) in certain deep machine algorithms (Lee et al., 2024b) by allocating 8-bits towards the exponent, as the larger 'Float32' format does. Broadly, an $n$-bit floating point datatype posses $n - 1$ possible distributions as $m \in [0, n-1]$, and even adopts the uniform distribution of the corresponding $n$-bit integer format when $m = n - 1$.

Figure 1 visualizes this advantage by showing the discrete value distribution of INT4 and FP4 under different FP formats. The bits allocation between the mantissa and exponent significantly influences the performance of quantization as depicted. While the flexibility of floating-point format benefits the quantization, improper FP format can result in sub-optimal performance (Shen et al., 2024). Hence, in the following section, we present our analysis of the DiT blocks and introduce our method, which adjusts the FP format when quantizing different DiT weights.

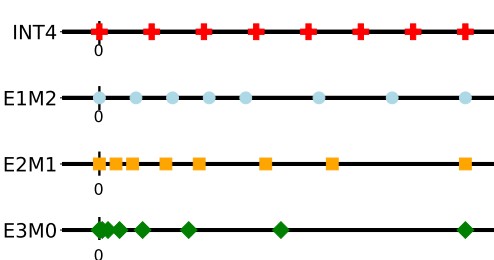

Figure 1: Value distributions for INT4 and three variants of FP4: E1M2, E2M1 and E3M0. Note that E0M3 is INT4. Observe how INT4 values are evenly distributed, while FP4 values cluster at the origin as the number of exponent (E) bits increases.

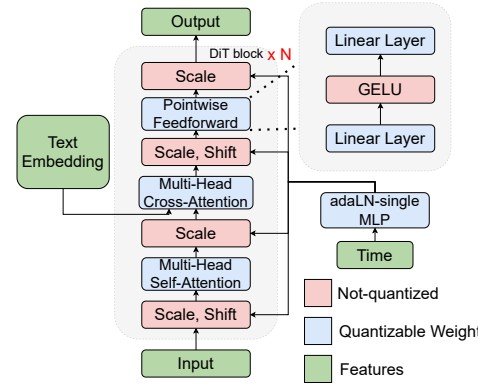

Figure 2: T2I DiT block diagram. In PixArt-$\alpha/\Sigma$, all DiT blocks share the same AdaLN-single MLP for time conditions. The scale and shift for layer normalization in DiT blocks depend on the embedding from AdaLN-single and the layer-specific training embedding. Colored blocks demarcate quantizable weight layers from activations and normalizations.

### 3.1.1 Optimized FP Formats in DiT Blocks.

Figure 2 illustrates the structure of a T2I DiT Block. In a DiT block, the Pointwise Feedforward is unique in that it consists of a non-linear GELU activation flanked by linear layer before and after. GELU, plotted in Figure 3 contains a sensitive region where the function returns a negative output. Interestingly, Reggiani et al., 2023 (Reggiani et al., 2023) show that focusing on this sensitive interval helps reduce the mean-squared error when approximating GELU using Look-Up Tables (LUTs) or breakpoints. Building on this insight, we apply denser floating point formats, e.g., E3M0, to the first pointwise linear layer. This allocates more values closer to zero, i.e., where the GELU sensitive interval lies, thereby enhancing the precision of the approximation. We further elaborate on this point in the appendix.

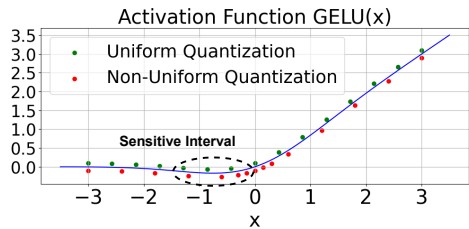

Figure 3: The GELU activation and its sensitive interval. With the same amount of discrete values, non-uniform quantization can better capture the sensitive interval.

## 3.2 AdaRound for FP

By default, quantization is rounding-to-nearest, e.g., Eq. 1. AdaRound (Nagel et al., 2020) show that rounding-to-nearest is not always optimal and instead apply second-order Taylor Expansion on the loss degradation from weight perturbation $\Delta \mathbf{w}$ caused by quantization:

$$E[\Delta L(\mathbf{w})] \approx \Delta \mathbf{w}^T \mathbf{g}^{(\mathbf{w})} + \frac{1}{2}\Delta \mathbf{w}^T \mathbf{H}^{(\mathbf{w})}\Delta \mathbf{w}. \tag{3}$$

The gradient term $\mathbf{g}^{(\mathbf{w})}$ is close to 0 as neural networks are trained to be converged. Hence, the loss degradation is determined by the Hessian matrix $\mathbf{H}^{(\mathbf{w})}$, which defines the interactions between different perturbed weights in terms of their joint impact on the task loss. The rounding-to-nearest is sub-optimal because it only considers the on-diagonal elements of $\mathbf{H}^{(\mathbf{w})}$. However, optimizing via a full Hessian matrix is infeasible because of its computational and memory complexity issues. To tackle these issues, the authors make assumptions such as each non-zero block in $\mathbf{H}^{(\mathbf{w})}$ corresponds to one layer, and then propose an objective function:

$$\arg\min_{\mathbf{V}} \|W\mathbf{x} - \widetilde{W}\mathbf{x}\|_F^2 + \lambda f_{\text{reg}}(\mathbf{V}), \tag{4}$$

The optimization objective is to minimize the Frobenius norm of the difference between the full-precision output $W\mathbf{x}$ and the quantized output $\widetilde{W}\mathbf{x}$ for each layer and $f_{\text{reg}}(\mathbf{V})$ is a differentiable regularizer to encourage the variable $\mathbf{V}$ to converge. BRECQ (Li et al., 2021) proposed a similar block-wise optimization objective that further advances the performance of weight reconstruction in PTQ. In detail, $\widetilde{W}$ is defined as follows:

$$\widetilde{W} = s \cdot \text{clip}\left(\left\lfloor\frac{W}{s}\right\rfloor + h(\mathbf{V}), min, max\right) \tag{5}$$

where $min$ and $max$ denotes the quantization threshold. $h(\mathbf{V})$ is the rectified sigmoid function proposed by (Louizos et al., 2017):

$$h(\mathbf{V}) = \text{clip}\left(\sigma(\mathbf{V})(\zeta - \gamma) + \gamma, 0, 1\right) \tag{6}$$

where $\sigma(\cdot)$ is the sigmoid function and $\zeta$ and $\gamma$ are fixed to 1.1 and -0.1. During optimization the value of $h(\mathbf{V})$ is continuous, while during inference its value will be set to 0 or 1 indicating rounding up or down.

### 3.2.1 Scale-aware AdaRound.

AdaRound has been widely adopted to improve the performance of quantized neural networks (Li et al., 2021; 2023) in low-bit settings like 4-bit weights. However, AdaRound assumes weight quantization to low-bit integer formats, like INT4 rather than low-bit FP formats (van Baalen et al., 2023), where non-uniform value distribution (Fig. 1) may introduce unique challenges.

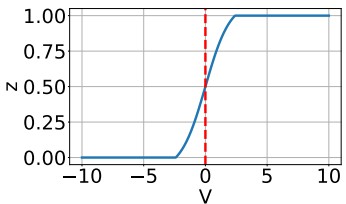 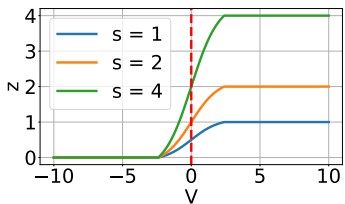 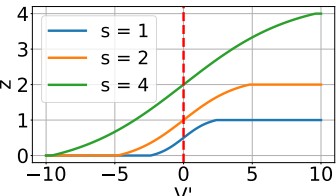

(a) AdaRound for Integers     (b) AdaRound Applied to Floats     (c) Scale-Aware AdaRound

Figure 4: (a) The binary gate function of INT AdaRound. All gates are identical because there is only one scale in INT quantization. (b) The binary gate functions of origin AdaRound on FP quantization. (c) The binary gate functions of scale-aware AdaRound. The red dashed line indicates the demarcation of rounding up (right) or down (left). Our scale-aware AdaRound normalizes the slope near the turning point, which stabilizes the optimization and helps improve the quantization performance.

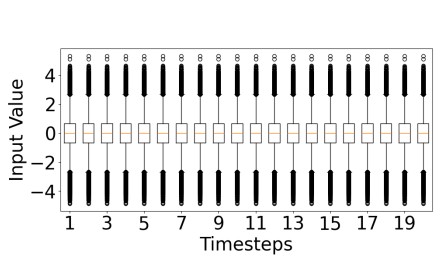 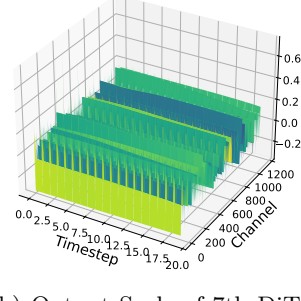 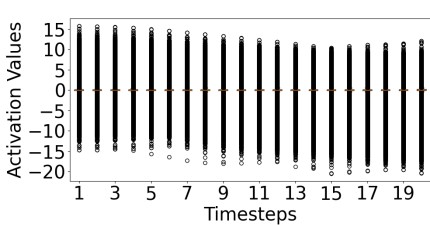

(a) DiT Input Value Box Plot     (b) Output Scale of 7th DiT Block     (c) 7th DiT Block Output Box Plot

Figure 5: (a) Different timestep input values for PixArt-$\alpha$ on 128 images sampled from MS-COCO. The input does not shrink progressively across timesteps like U-Net DM. (b) The time-embedded scale for the output of the 7th DiT block's FeedForward. It is almost constant across timesteps. (c) The output of the 7th DiT block. Its range tends to remain constant but shifts as a function of time.

Specifically, we identify that the original INT-based AdaRound assumes the scale $s$ is consistent across different quantized values. However, this does not hold for FPQ, where there are $2^E$ scales. Therefore, we propose scale-aware AdaRound which improves the performance and leads to faster convergence.

Our scale-aware AdaRound inherits Equation 4 as the learning objective because reducing the layer-wise and block-wise quantization error is the common goal of FPQ and INT quantization. Differently, We modified the $\widetilde{W}$ as:

$$\widetilde{W} = s \cdot \text{clip}\left(\left\lfloor \frac{W}{s} \right\rfloor + h'(\mathbf{V}'), min, max\right) \tag{7}$$

$$h'(\mathbf{V}') = \text{clip}\left(\sigma(\frac{\mathbf{V}'}{s})(\zeta - \gamma) + \gamma, 0, 1\right) \tag{8}$$

where $h'(\cdot)$ is the scale-aware rectified sigmoid function and $V'$ is a new continuous variable we optimized over.

The rectified sigmoid function functions as binary gates that control the rounding of weights. Specifically, the gate function $z = s \cdot h(V)$ is optimized according to Equation 4. Figure 4a shows those binary gates in INT AdaRound. The gates are equivalent across all the weights, which matches the even distribution of INT quantization. In Figure 4b, we show the origin AdaRound's binary gates under different scales. The gates' slope depends on their scale, which causes imbalanced update during the gradient descent. In Figure 4c, we show the binary gates of our scale-aware AdaRound. In contrast, the gates' slope is normalized to the same level, which makes the weight reconstruction much more stable and thus aids the quantization. For the mathematical proof of our scale-aware AdaRound, please refer to the appendix.

### 3.3 Token-wise Activation Quantization

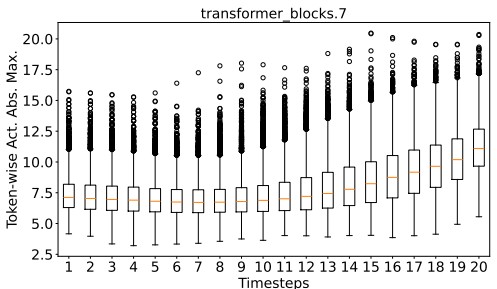

Figure 6: The distribution of the absolute maximum for each token's activation among 4096 tokens in the PixArt-$\alpha$ model. The distribution demonstrates a strong patch dependency in the DiT activation.

Prior DM PTQ approaches (He et al., 2024; Huang et al., 2024) use a calibration dataset to learn temporally-aware (He et al., 2023) activation quantization scales. This approach is predicated on knowledge of how U-Net activation distributions change as a function of denoising timesteps, i.e., activation ranges taper-off towards the end of the denoising process (Shang et al., 2023b; Li et al., 2023).

In contrast, we show that this assumption does not hold for DiT models. First, we collect input activations of PixArt-$\alpha$ across 20 timesteps, revealing that the activation range remains stable over time, as shown in Figure 5a. We then analyze the Adaptive Layernorm (AdaLN) (Perez et al., 2018) in the PixArt DiT model in Figure 2. Since the feed-forward scale directly influences the DiT block output range, we visualize the feed-forward scale in the first layer in Figure 5b and the output of the 7th DiT block in Figure 5c. These figures demonstrate that the value of activations is primarily controlled by channels opposed to timesteps, and that the width of the activation range tends to remain constant, but shifts as a function of time.

Further, we plot the token-wise activation range in Figure 6. This plot visualizes the absolute maximum activation of each image patch (token) across time. The results indicate that the activation range varies significantly, even among tokens within the same timestep.

Recent works by Microsoft (Yao et al., 2022; Wu et al., 2023b) propose an online token-wise activation quantization that yields superior results when quantizing transformer activations. Table 1 applies this technique to the DiT scenario. Specifically, we apply simple min-max quantization to reduce weight precision of PixArt-$\alpha$ to 4-bits (W4), then consider 8-bit (A8) and 6-bit (A6) activation quantization. In both scenarios, we observe CLIP performance that is closer to the full precision model using token-wise activation quantization as opposed to the traditional, temporally-aware scale calibration technique which is designed for U-Nets. As such, we consider token-wise activation quantization throughout the remainder of this work by substituting it into U-Net baselines like Q-Diffusion (Li et al., 2023) and TFMQ-DM (Huang et al., 2024).

## 4 Results

In this section we conduct experiments to verify the efficacy of FP4DiT. We elaborate on our experimental setup and then compare FP4DiT to several baselines approaches to highlight its competitiveness in terms of quantitative metrics and qualitative image generation output. Specifically, we consider three text-to-image (T2I) models: PixArt-$\alpha$ (Chen et al., 2024b), PixArt-$\Sigma$ (Chen et al., 2024a) and Hunyuan (Li et al., 2024). We also conduct several ablation studies to verify the components of our method. Finally, we report several hardware metrics tabulating the cost savings and throughput of FP4DiT.

### 4.1 Experimental Settings

We use the HuggingFace Diffusers library (von Platen et al., 2022) to instantiate the base DiT models in W16A16 bit-precision and consider the default values for

| Method | Precision | CLIP ↑ |
|---|---|---|
| No Quantization | W16A16 | 0.3075 |
| Temporally-Aware Act. Quant. | W4A6 | 0.2012 |
| Token-wise Act. Quant. | W4A6 | 0.3036 |
| Temporally-Aware Act. Calib. | W4A8 | 0.2410 |
| Token-wise Act. Quant. | W4A8 | 0.3120 |

Table 1: CLIP score (Hessel et al., 2021) reported when quantizing PixArt-$\alpha$ to 4-bit weights (W4) and 8 or 6-bit activations (A8 and A6) using the online token-wise method and temporarally-aware scale calibration. Specifically, we generate 1k images per configuration using COCO (Lin et al., 2014a) prompts and compare against the validation set. Higher CLIP is better.

| Benchmark | | | HPSv2 | | | | | MS-COCO | | |
|---|---|---|---|---|---|---|---|---|---|---|
| Model | Method | Precision | Animation↑ | Concept-art↑ | Painting↑ | Photo↑ | Average↑ | FID ↓ | CLIP↑ | IR↑ |
| PixArt-α | Full Precision | W16A16 | 32.56 | 31.06 | 30.76 | 29.67 | 31.01 | 34.05 | 0.3075 | 0.7037 |
| | Q-Diffusion | W4A8 | 24.18 | 23.43 | 22.91 | 22.15 | 23.17 | 52.20 | 0.3017 | 0.1855 |
| | TFMQ-DM | W4A8 | 27.88 | 26.03 | 25.14 | 24.91 | 25.99 | 64.73 | *0.3066* | 0.4706 |
| | ViDiT-Q | W4A8 | 17.93 | 17.35 | 16.81 | 17.59 | 17.42 | 39.11 | 0.2900 | 0.4716 |
| | Q-DiT | W4A8 | *28.49* | **26.91** | *27.55* | *26.56* | *27.38* | *36.17* | 0.3062 | *0.4979* |
| | FP4DiT (ours) | W4A8 | **28.63** | *26.79* | **27.59** | **26.72** | **27.43** | **30.37** | **0.3076** | **0.5509** |
| | Q-Diffusion | W4A6 | 12.63 | 13.39 | 13.32 | 10.65 | 12.50 | 70.96 | 0.2868 | -0.1825 |
| | TFMQ-DM | W4A6 | 24.02 | **23.36** | 22.72 | *23.04* | 23.29 | 90.09 | *0.3015* | 0.1299 |
| | ViDiT-Q | W4A6 | 16.71 | 16.28 | 16.23 | 16.56 | 16.44 | 56.54 | 0.2827 | 0.1296 |
| | Q-DiT | W4A6 | **24.66** | 22.59 | *23.29* | *23.04* | *23.40* | *43.91* | *0.3015* | *0.1323* |
| | FP4DiT (ours) | W4A6 | *24.57* | *23.08* | **23.30** | **23.26** | **23.55** | **40.61** | **0.3031** | **0.1353** |
| PixArt-Σ | Full Precision | W16A16 | 33.07 | 31.58 | 31.54 | 30.49 | 31.67 | 36.94 | 0.3139 | 0.9223 |
| | Q-Diffusion | W4A8 | 27.30 | 26.11 | 26.06 | **25.06** | 26.13 | 36.89 | *0.3050* | 0.2285 |
| | TFMQ-DM | W4A8 | 24.95 | 23.59 | 23.19 | 22.73 | 23.62 | 62.98 | 0.2975 | 0.1047 |
| | ViDiT-Q | W4A8 | 27.10 | **26.45** | **26.24** | 24.49 | 26.07 | 37.17 | 0.2562 | *0.2832* |
| | Q-DiT | W4A8 | *27.74* | 26.23 | 26.05 | *24.67* | *26.17* | *32.03* | 0.3048 | 0.2679 |
| | FP4DiT (ours) | W4A8 | **27.95** | *26.29* | *26.16* | *24.67* | **26.27** | **31.58** | **0.3064** | **0.2927** |
| | Q-Diffusion | W4A6 | 24.07 | 22.42 | 22.47 | 21.72 | 22.67 | 74.83 | *0.3027* | 0.0828 |
| | TFMQ-DM | W4A6 | 19.30 | 18.46 | 18.85 | 17.50 | 18.53 | 154.13 | 0.2346 | -0.2281 |
| | ViDiT-Q | W4A6 | 24.84 | 23.23 | 23.48 | 21.94 | 23.37 | 87.47 | 0.2425 | -0.1156 |
| | Q-DiT | W4A6 | *25.67* | *23.57* | *23.60* | *22.64* | *23.87* | *73.20* | 0.2894 | *0.1614* |
| | FP4DiT (ours) | W4A6 | **26.91** | **25.55** | **25.22** | **23.93** | **25.40** | **42.21** | **0.3040** | **0.2085** |
| Hunyuan | Full Precision | W16A16 | 33.72 | 31.84 | 31.52 | 31.24 | 32.08 | 59.08 | 0.3102 | 0.8701 |
| | Q-Diffusion | W4A8 | 26.00 | 24.74 | 24.84 | 24.26 | 24.96 | 82.43 | 0.3006 | 0.6874 |
| | TFMQ-DM | W4A8 | 28.77 | *27.63* | **27.67** | 26.15 | *27.56* | 89.39 | 0.3075 | 0.6873 |
| | ViDiT-Q | W4A8 | 28.74 | 26.79 | 26.49 | **27.20** | 27.31 | *82.22* | *0.3099* | 0.7423 |
| | Q-DiT | W4A8 | *28.91* | 27.27 | 27.15 | 26.72 | 27.52 | 85.33 | 0.3088 | *0.7539* |
| | FP4DiT (ours) | W4A8 | **28.96** | **27.75** | *27.47* | *26.94* | **27.78** | **81.94** | **0.3102** | **0.7669** |
| | Q-Diffusion | W4A6 | 14.90 | 14.83 | 15.32 | 14.24 | 14.83 | *255.69* | 0.2277 | -1.9236 |
| | TFMQ-DM | W4A6 | 13.73 | 13.31 | 13.16 | **14.51** | 13.68 | 258.02 | 0.2520 | -1.9007 |
| | ViDiT-Q | W4A6 | *15.11* | **15.20** | *15.41* | 14.17 | *14.97* | 265.52 | *0.2559* | -1.8123 |
| | Q-DiT | W4A6 | 13.41 | 13.15 | 13.48 | 12.94 | 13.24 | 262.31 | 0.2312 | **-1.1822** |
| | FP4DiT (ours) | W4A6 | **15.23** | *14.94* | **15.57** | *14.27* | **15.00** | **255.06** | **0.2562** | *-1.7880* |

Table 2: Quantitative evaluation results for PixArt-α, PixArt-Σ and Hunyuan in terms of HPSv2, FID, CLIP and ImageReward (IR) score. Specifically, for each configuration, we generate 10k images (5k for Hunyuan) using COCO 2014 validation set prompts. Best and second best results in **bold** and *italics*, respectively.

inference parameters like number of denoising steps and classifier-free guidance (CFG) scale. We quantize weights to 4-bit precision FP format. Specifically, we set the weight format for the first linear layer in each pointwise feed-forward to be E3M0. We quantize all other weights to E2M1 for PixArt-α and Hunyuan, and E1M2 for PixArt-Σ. Further details on this decision are provided in Session 4.3.1. Finally, our weight quantization is group-wise (Frantar et al., 2022; Park et al., 2022) along the output channel dimension with a group size of 128.

We perform weight quantization calibration using our scale-aware AdaRound and BRECQ. Weight calibration requires a small amount of calibration data. We use 128 (64 for Hunyuan) prompts from the MS-COCO 2014 train (Lin et al., 2014a) dataset and calibrate for 2.5k iterations per DiT block or layer. Next, we perform activation quantization to 8 or 6-bit precision using min-max token-wise quantization from Zero-Quant (Yao et al., 2022; Wu et al., 2023b). We provide further hyperparameter details in the appendix.

## 4.2 Main Results

In our experiment, we consider four baseline approaches: Q-Diffusion (Li et al., 2023), TFMQ-DM (Huang et al., 2024), ViDiT-Q (Zhao et al., 2024) and Q-DiT (Chen et al., 2025a). Note that while Q-Diffusion and TFMQ-DM are originally designed for U-Nets, we modify these approaches to use the same online token-wise activation quantization as FP4DiT per Table 1, while ViDiT-Q and Q-DiT uses this mechanism by default.

Further, note that ViDiT-Q uses mix-precision meaning some of their layers are not quantized to 4-bits. For the sake of convenience, we use their mix-precision as a W4 baseline to conduct our experiments.

We generate $512 \times 512$ resolution images using PixArt-$\alpha$ and $1024 \times 1024$ for PixArt-$\Sigma$ and Hunyuan. For evaluation, we primarily consider the Human Preference Score v2 (HPSv2) (Wu et al., 2023a) benchmark and MS-COCO 2014 (Lin et al., 2014a) validation set. Specifically, HPSv2 considers four image categories: animation, concept-art, painting and photography, and estimates the human preference score of an image generated using a prompt with respect to one category. Each category contains 800 prompts, requiring 3.2k images to be generated to fully evaluate. The final HPSv2 score is the average estimated human preference across all four categories. Additionally, for COCO 2014, we measure the FID (Heusel et al., 2017), CLIP (Hessel et al., 2021) score and average ImageReward (IR) (Xu et al., 2023) score using prompts from the validation set.

Table 2 compares our method with the stated baseline approaches at the W4A8 and W4A6 precision levels. FP4DiT consistently outperforms all other methods across three different base models at every precision level in terms of each performance metric. On HPSv2, FP4DiT achieves higher performance across every model and bit-precision combination. Although other approaches may achieve slightly higher performance on an individual category, these gaps are small, while FP4DiT leads in terms of FID, and CLIP on the COCO 2014 validation set. Finally, we observe a modest amount of IR degradation for PixArt-$\alpha$ and Hunyuan compared to the W16A16 model, however this is most pronounced for PixArt-$\Sigma$ where all quantization methods suffer a substantial IR score loss.

We believe this degradation is two-fold. First, PixArt-$\Sigma$ generates larger images, which means there is more pixel space to introduce noise. Second, IR estimates how much a human would prefer an image given the prompt, as opposed to CLIP which measures prompt adherence by comparing the latent embeddings of prompt and image, meaning the errors introduced by quantization are likely to have a greater impact. Nevertheless, FP4DiT achieves the highest IR performance in most of the model and precision settings.

Next, we compare performance across activation bit-precision. For the PixArt-series DiT models, A6 precision can cause substantial performance degradation for some baselines, while FP4DiT still maintains high performance. This is specifically noteworthy on PixArt-$\Sigma$ as baseline approaches suffer a loss in terms of either FID and/or CLIP performance at A6 compared to A8, but this is not the case for FP4DiT. For the Hunyuan DiT model, A6 quantization remains highly challenging across the board, however, our method still achives the best results at this level except the IR score. Overall though, Table 2 demonstrates the efficacy of FP4DiT in term of quantitative T2I compared to prominent baseline approaches.

Next, Figure 7 provides qualitative image samples on the PixArt models. Note the higher quality of the images generated by FP4DiT at both the A8 and A6 levels. Specifically, on PixArt-$\alpha$, the puppies have more realistic detail and the generated image more closely aligns with the W16A16 model. This is especially true for the PixArt-$\Sigma$ sample images, where FP4DiT accurately captures the intricate details of the portrait of a middle-aged Asian woman, including the fractured porcelain and paint splatters. In contrast, the other quantization methods either introduce significant artifacts, lose fine details, or fail to accurately represent the prompt's content, further highlighting the robustness and superior performance of the FP4DiT approach.

Further, Figure 8 provides image results on Hunyuan, where we again note the detail present in the FP4DiT image, while the baseline approaches are much noisier and have yellow backgrounds which are not prompt-adherent. Finally, additional visualization results can be found in the appendix.

### 4.2.1 User Preference Study

To further verify the utility of our method, we conduct several human user preference studies to qualitatively compare FP4DiT to existing baseline approaches. Specifically, for each human user preference study, we have a set of prompts (i.e., introduced in (Chen et al., 2024b)) as well as quantized variants of a single model (e.g., Hunyuan-DiT quantized to W4A8 using ViDiT-Q, Q-DiT and FP4DiT). Each quantized model generates one image per prompt. For each prompt, we solicit the opinion of a human participant, by presenting them the set of images produced using the prompt by different methods, and ask them to select which image they prefer, in terms of perceived prompt adherence and general visual quality. We then tally up the number of

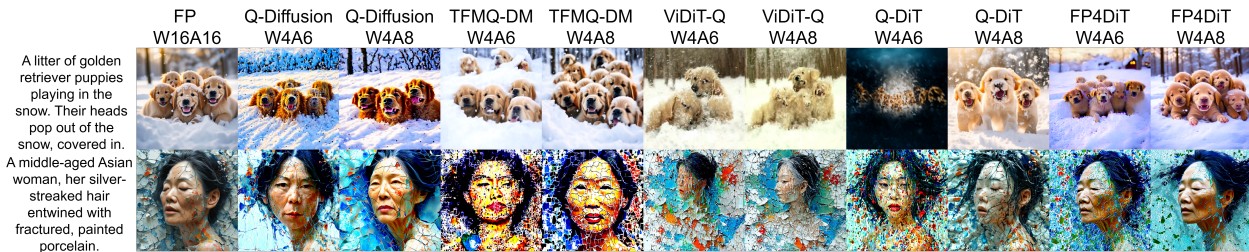

Figure 7: PixArt-$\alpha$ (up) and PixArt-$\Sigma$ (down) images and comparison between FP4DiT and related work. Best viewed in color and zoomed in.

Figure 8: Hunyuan image comparison. Note the details like 'white background' and 'detailed hair texture' for FP4DiT. Best viewed in color.

| Model | Method | Prec. | Preference(%) ↑ |
|---|---|---|---|
| PixArt-$\Sigma$ | ViDiT-Q | W4A6 | 17.09 |
| | Q-DiT | W4A6 | 30.77 |
| | FP4DiT | W4A6 | **52.14** |
| Hunyuan | ViDiT-Q | W4A8 | 19.17 |
| | Q-DiT | W4A8 | 35.83 |
| | FP4DiT | W4A8 | **45.00** |

| Model | Method | Prec. | Preference(%) ↑ |
|---|---|---|---|
| Hunyuan | Q-Diffusion | W4A8 | 6.58 |
| | TFMQ-DM | W4A8 | 36.84 |
| | FP4DiT | W4A8 | **56.58** |
| | Q-Diffusion | W4A6 | 32.84 |
| | TFMQ-DM | W4A6 | 28.36 |
| | FP4DiT | W4A6 | **38.81** |

Table 3: DiT PTQ user preference study between FP4DiT, ViDiT-Q, and Q-DiT on PixArt-$\Sigma$ W4A6 precision and Hunyuan-DiT W4A8 precision.

Table 4: AdaRound/BRECQ user preference study between FP4DiT, Q-Diffusion and TFMQ-DM on Hunyuan DiT with W4A8 and W4A6 quantization.

votes each method obtains and compute how often it is selected compared to its competitors, e.g., preference score (%), where higher is better.

Our first test focused on comparing FP4DiT to other methods explicitly designed for DiT PTQ: ViDiT-Q and Q-DiT. This study consists of 120 prompts and 17 human participants. We compare generated image quality for PixArt-$\Sigma$ at W4A6 precision and Hunyuan at W4A8 precision. Table 3 reports our findings. FP4DiT is preferred over 50% of the time for PixArt-$\Sigma$ W4A6 precision level, followed by Q-DiT and then ViDiT-Q. This result aligns with Table 2 where FP4DiT achieves the best result and Q-DiT outperforms ViDiT-Q in this setting. For Hunyuan at W4A8 the preference vote share for baseline methods grow, but they do not overtake FP4DiT, which demonstrates the veracity of our approach.

Next, we conduct an additional study comparing FP4DiT to baseline approaches originally designed with U-Net DMs in mind but which utilize advanced PTQ via AdaRound and BRECQ: Q-Diffusion and TFMQ-DM. Specifically, we consider 75 random prompts and 15 human participants. The baseline DM is Hunyuan quantized to either W4A8 or W4A6 precision. Table 4 reports our findings. At A8 precision FP4DiT clearly outperforms the other methods with a majority of the images being favored, while it also obtains almost 40% of the votes at A6 precision.

### 4.3 Ablation Studies

We conduct ablation studies on the PixArt-$\alpha$ model to verify the contribution of each component of FP4DiT. The experiment settings are consistent with Section 4.2 unless specified. Additional ablation studies can be found in the appendix.

#### 4.3.1 Effect of FP Format.

Recall the mixed format strategy for FPQ in FP4DiT: we apply E3M0, which allocates more value closer to the GELU sensitive interval, to the first pointwise linear layer and use a unified FP format from E2M1 and E1M2 for the rest layers. To choose a better one between E2M1 and E1M2 while avoiding data leakage, instead of quantizing FP4DiT with two formats and selecting the better one based on the FID and CLIP, we only employ the basic min-max quantization scheme (without AdaRound technique thereby avoiding the use of calibration data) and leave the activation un-quantize (e.g. W4A16). We use the PixArt prompts to generate images and apply user preference studies to determine the format.

Figure 9 shows the visualization results of E2M1 and E1M2 FPQ. For the PixArt-$\alpha$, E2M1 demonstrates better suitability, as the cactus in E1M2 loses its texture, resulting in a mismatch between the image and the prompt. For PixArt-$\Sigma$ and Hunyuan, inappropriate FP format causes noise on the generated image, leading to suboptimal performance. In conclusion, it is straightforward and unambiguous to determine the unified FP format based on these visualization results.

#### 4.3.2 Effect of Weight Quantization

To evaluate the effectiveness of our weight quantization method, we perform an ablation study on weight-only quantization, e.g. W4A16. As depicted in Table 5, our method progressively improves the weight quantization: Initially, directly applying FPQ to Q-Diffusion results in a significant degradation. Our research then reveals that group quantization is necessary for FPQ. Notably, using the original AdaRound on top of FPQ impedes its effectiveness. Subsequently, our sensitive-aware mixed format FPQ (E3M0 in pointwise linear) further improves the post-quantization performance. Eventually, our scale-aware AdaRound advances the boundaries of optimal performance by considering the multi-scale nature of FP weight reconstruction.

#### 4.3.3 Effect of Scale-Aware AdaRound

To further verify our scale-aware AdaRound for FP quantization, we compare our scale-aware AdaRound to the origin AdaRound with INT quantization in the W4A16

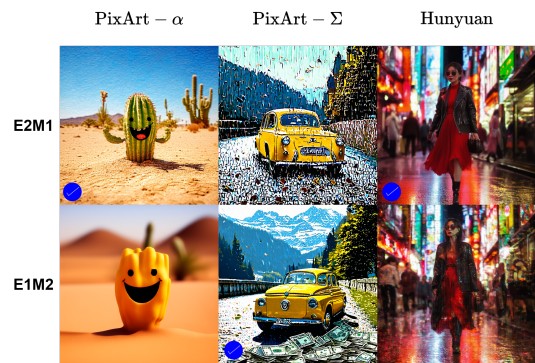

Figure 9: E2M1 (up) and E1M2 (down) min-max FPQ visualization results on PixArt and Hunyuan DiT. E2M1 is preferred by PixArt-$\alpha$ and Hunyuan, while E1M2 is preferred by PixArt-$\Sigma$.

| Method | Precision | HPSv2 ↑ |
|---|---|---|
| Full Precision | W16A16 | 31.01 |
| Q-Diffusion | W4A16 | 25.22 |
| Q-Diffusion-FPQ | W4A16 | 8.78 |
| + Group Quant | W4A16 | 25.05 |
| + Scale-Aware AdaRound | W4A16 | 26.44 |
| + Sensitive-aware FF Quant. | W4A16 | 28.21 |

Table 5: The effect of different methods proposed for weight quantization on W4A16 PixArt-$\alpha$.

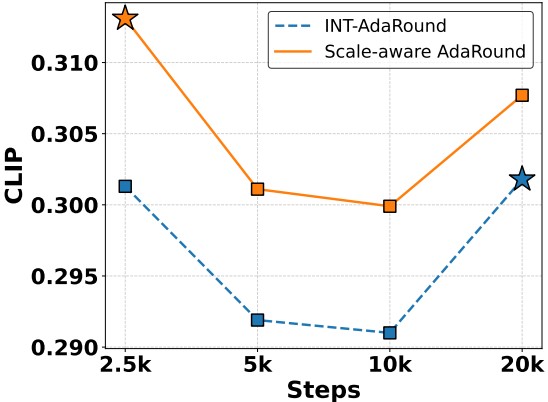

Figure 10: The performance of different AdaRound methods with different calibration budgets on W4A16 PixArt-$\alpha$ quantization generating 1k images. The stars indicate the optimal budget for INT and scale-aware AdaRound.

| Layer | W4A8-FP (ms) | W4A6-FP (ms) | W4A8-INT (ms) | W16A16 (ms) |
|---|---|---|---|---|
| Feedforward Layer 1 | 303.42 (1.54×) | 313.54 (1.49×) | 302.56 (1.54×) | 465.99 |
| Feedforward Layer 2 | 313.31 (1.50×) | 318.67 (1.47×) | 312.80 (1.50×) | 469.91 |
| Self-Attention QKV Proj. | 78.33 (1.51×) | 79.94 (1.48×) | 78.16 (1.51×) | 118.28 |

Table 6: The latency and the speedup of different linear layers in PixArt-$\alpha$/$\Sigma$ under different quantization precision generating a $1024\times1024$ image, e.g., batch size of 2, 4096 tokens, each with an embedding dimension of 1152. Result measured on RTX 5080 GPU using CUDA 12.8.

| GPU | INT8 | FP8 | FP6 | FP4 |
|---|---|---|---|---|
| RTX 4090 (NVIDIA, 2024a) | 660.6 TOPS | 660.6 TFLOPS | – | – |
| H100 (NVIDIA, 2024c) | 1979 TOPS | 1979 TFLOPS | – | – |
| RTX 5090 (NVIDIA, 2024a) | 838 TOPS | 838 TFLOPS | 838 TFLOPS | 1676 TFLOPS |
| HGX B100 (NVIDIA, 2024b) | 56 POPS | 56 PFLOPS | 56 PFLOPS | 112 PFLOPS |
| HGX B200 (NVIDIA, 2024b) | 72 POPS | 72 PFLOPS | 72 PFLOPS | 144 PFLOPS |

Table 7: GPU throughput rates for different low-bit datatype formats. Horizontal line demarcates older Ada Lovelace/Hopper GPUs from state-of-the-art Blackwell series. Older series do not support FP6 and FP4.

setting. Note that the calibration budget is crucial for
the performance of weight reconstruction. BRECQ (Li et al., 2021) uses 20k as default and this setting is inherited by prior DM quantization research like Q-Diffusion and TFMQ-DM. Thus, we configured calibration budgets at {2.5k, 5k, 10k, 20k} to ensure a fair comparison. Figure 10 outlines the CLIP of different AdaRound methods on PixArt-$\alpha$ under different calibration budgets. Scale-aware AdaRound achieving optimal budget with 8 times fewer calibration steps than INT AdaRound, highlights its effectiveness in reducing calibration costs without compromising reconstruction quality.

## 4.4 Hardware Cost Comparison

Finally, we measure the hardware latency impact of FPQ compared to more traditional INT quantization and the W16A16. Specifically, we consider the quantization latency cost to execute some of the common, yet costlier weight/activation operations in a DiT, namely the self-attention mechanism and feedforward module. Specifically, the linear layers corresponding to the self-attention mechanism (Q, K, V and output projection) all share the same weight/activation dimensions, while the feedforward consists of two linear layers which expand and then contract the token/patch embedding dimension, respectively. Additionally, weights and activations may not share the same bit precision, imposing additional dequantization cost.

We develop a CUDA 12.8 kernel and measure hardware latency on an Nvidia 5080, a Blackwell GPU which supports low-bit FP formats. We consider the self-attention weight/activation dimensions found in PixArt-$\alpha$/$\Sigma$ when generating an $1024 \times 1024$ image. Table 6 reports our findings. The result indicates that, despite the general expectation that floating-point computations are more demanding than integer operations, the latency results are nearly identical across all tested quantization precisions. Specifically, although A6 precision imposes some additional overhead compared to A8 (either for FP or INT), this is quite minor, leading to an overall speedup around 1.5x compared to W16A16, much like the popular W4A8 integer quantization (Li et al., 2023; Zhao et al., 2024; Chen et al., 2025a).

Leveraging this finding, our FPQ method can achieve superior performance without sacrificing computational efficiency. In fact, this latency results align with the stated computation rate of different INT and FP formats across different Nvidia GPUs, which we report in Table 7. Specifically, GPUs can handle INT and FP quantization with similar computational throughput, which ensures that FP-based quantization does not introduce additional computational overhead. Lastly, although FP6 shares the same compute throughput as 8-bit formats (see Table 7), it brings memory saving (25% smaller tensor) which is essential for low-bit quantization, as transformers often become memory-bound (Xia et al., 2024) during token generation. As a result, FP6 enhances DiT's inference efficiency.

| Model | Precision | Model Size (MB)↓ | TBOPs↓ |
|---|---|---|---|
| PixArt-$\alpha$ | W16A16 | 610.86 | 35.72 |
| | W8A8 | 305.53 | 8.938 |
| | W4A8 | 152.87 | 4.474 |
| | W4A6 | 152.87 | 3.358 |
| | W4A8-G128 (ours) | 158.59 | 4.474 |
| | W4A6-G128 (ours) | 158.59 | 3.358 |

Table 8: The comparison of model size and Bit-Ops of different Precision on PixArt-$\alpha$. G128 denotes group-wise weight quantization with a group size of 128. Lower is better.

We also compare the hardware cost of the quantized FP4DiT model with the full-precision model in terms of memory and energy consumption using two metrics: model size and Bit-Ops (BOPs) (He et al., 2024). Specifically, model size refers to the disk space required to store the model checkpoint weights and scales, while BOPs is a quantization-aware extension of the MACs (Chu et al., 2022; Mills et al., 2023) metric which measures the compute cost of a neural network forward pass. As shown in Table 8, group-wise weight quantizationintroduces only moderate overhead; nonetheless, our method still substantially reduces the model size and BOPs.

## 5 Conclusion

In this paper, we propose FP4DiT, a PTQ method that achieves W4A6 and W4A8 quantization on T2I DiT using FPQ. We use a mixed FP formats strategy based on the special structure of DiT and propose scale-aware AdaRound to enhance the weight quantization for FPQ. We analyze the difference between the activation of U-Net DM and DiT and apply token-wise online activation quantization based on the findings. Our experiments demonstrate that FP4DiT achieves performance improvement in terms of quantative benchmarks such as CLIP and ImageReward scores on the MS-COCO dataset and HPSv2 benchmark, while also obtaining qualitative visualization improvement at minimal hardware cost.

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

# A Appendix

## A.1 Proof for Scale-Aware AdaRound

AdaRound (Nagel et al., 2020) and BRECQ (Li et al., 2021) use gradient descent to update the rounding mask. According to Eqs. 4, 5 and 6, ignoring the regularizer $f_{\text{reg}}(\mathbf{V})$, we have following theorem which provides theoretical evidence that the origin AdaRound is imbalance across different scales $s$.

**Theorem 1.** *Let $s$ be the quantization scale corresponding to the rounding mask $V$. Then for gradient descent, given as $\mathbf{V}_{n+1} = \mathbf{V}_n - \alpha\nabla F(\mathbf{V}_n)$, the subtraction $\nabla F(\mathbf{V}_n)$ is dependent on the scalar $s$.*

*Proof.* Refers to Eqs. 4, 5 and gives:

$$\nabla F(\mathbf{V}_n) = \nabla_{\widetilde{W}}\|W\mathbf{x} - \widetilde{W}\mathbf{x}\|_F^2 \cdot s \cdot \frac{\partial h(\mathbf{V}_n)}{\partial \mathbf{V}_n} \tag{9}$$

Refers to Eq. 6 and gives:

$$\frac{\partial h(\mathbf{V})}{\partial \mathbf{V}} = (\zeta - \gamma) \cdot \sigma(\mathbf{V}) \cdot (1 - \sigma(\mathbf{V})) \tag{10}$$

Therefore, combining Eq. 9 and 10, the subtraction $\nabla F(\mathbf{V}_n)$ is scaled by $s$. □

This theorem indicates that the imbalance gradient descent occurs if applying origin AdaRound to FPQ, as shown in Figure 4b.

In the section 3, we propose a scale-aware version of AdaRound in Eq. 7 and 8. Ignoring the regularizer $f_{\text{reg}}$, we have the following theorem which provides theoretical evidence that the scale-aware AdaRound has balanced gradient descent's update across different scales $s$.

**Theorem 2.** *Let $s$ be the quantization scale corresponding to the rounding mask $V'$. Then for gradient descent, given as $\mathbf{V}'_{n+1} = \mathbf{V}'_n - \alpha\nabla F(\mathbf{V}'_n)$, the subtraction $\nabla F(\mathbf{V}'_n)$ is independent of the scalar $s$.*

*Proof.* The learning objective does not change for scale-aware AdaRound. Hence, from Eq. 4 and 7, we give:

$$\nabla F(\mathbf{V}'_n) = \nabla_{\widetilde{W}}\|W\mathbf{x} - \widetilde{W}\mathbf{x}\|_F^2 \cdot s \cdot \frac{\partial h'(\mathbf{V}'_n)}{\partial \mathbf{V}'_n} \tag{11}$$

Refer to Eq. 8 and give:

$$\frac{\partial h'(\mathbf{V}')}{\partial \mathbf{V}'} = \frac{(\zeta - \gamma)}{s} \cdot \sigma(\mathbf{V}') \cdot (1 - \sigma(\mathbf{V}')) \tag{12}$$

Combining Eq. 11 and 12, we get:

$$\nabla F(\mathbf{V}'_n) = \nabla_{\widetilde{W}}\|W\mathbf{x} - \widetilde{W}\mathbf{x}\|_F^2 \cdot (\zeta - \gamma) \cdot \sigma(\mathbf{V}') \cdot (1 - \sigma(\mathbf{V}'))$$

This result shows that subtraction $\nabla F(\mathbf{V}'_n)$ is independent of scale $s$. □

Therefore, the gradient descent is balanced and normalized among different scale $s$ with our scale-aware AdaRound, as depicted in Figure 4c.

## A.2 Additional Ablation Studies

In addition to the ablation studies in the main text, we include multiple ablation studies here to further demonstrate the design of FP4DiT.

### A.2.1 Effect of Group Size.

---

**Algorithm 1** MinMax Quantization for FP Format

---

**Require:** Full-precision array $A_{\text{FP}}$, number of bits $n$, exponent bits $n_e$, mantissa bits $n_m$, clipping value $maxval$

1: $A_{\text{abs}} \leftarrow \text{abs}(A_{\text{FP}})$
2: $bias \leftarrow 2^{n_e} - \log_2(A_{\text{abs}}) + \log_2(2 - 2^{-n_m}) - 1$
3: $A_{\text{clip}} \leftarrow \min(\max(A_{\text{FP}}, -maxval), maxval)$
4: $S_{\text{log}} \leftarrow \text{clamp}\big(\lfloor \log_2(\text{abs}(A_{\text{clip}})) \rfloor + bias, 1\big)$
5: $S \leftarrow 2.0^{(S_{\text{log}} - n_m - bias)}$
6: $result \leftarrow \text{round-to-nearest}(A_{\text{clip}}/S) \times S$
7: **return** $result$

---

We compare the group-wise weight quantization result with different group sizes in Table 9. Decreasing the group size $g$ for weight quantization consistently improves the quantization performance down to $g = 128$. According to Park et al. (2022), group size $g$ such as 128 can result in substantial improvement while maintaining low latency.

| Model | Group Size | FID ↓ |
|---|---|---|
| PixArt-$\alpha$ | 128 | 96.58 |
| | 256 | 100.60 |
| | 512 | 113.78 |
| | 1024 | 174.38 |

Table 9: The quantization results for different group sizes with PixArt-$\alpha$.

### A.2.2 Effect of Calibration Dataset

We investigate FP4DiT's effectiveness under different calibration dataset sizes $N$. Table 10 reports the HPSv2 score of the W4A8 PixArt-$\alpha$ quantized by our method with $N = 64, 128, 256$, and $512$. The results show steady improvements as $N$ increases, with performance saturating beyond $N = 256$. In particular, FP4DiT achieves a strong score of 27.43 with only 128 samples, and further gains to 28.21 at $N = 256$, while larger calibration does not bring additional benefits. This demonstrates that FP4DiT requires only a relatively small calibration set to achieve near-optimal performance, making it both efficient and effective compared to existing approaches.

| N=64 | N=64 | N=64 | N=64 |
|---|---|---|---|
| 25.09 | 27.43 | 28.21 | 28.20 |

Table 10: The quantization results for different sizes of calibration dataset with PixArt-$\alpha$.

### A.2.3 Classifier-Free Guidance (CFG)

| Precision | CFG=3.0 | CFG=4.5 | CFG=7.0 |
|---|---|---|---|
| W4A8 | 27.22 | 27.43 | 26.74 |
| Full Precision | 30.97 | 31.01 | 31.26 |

Table 11: The quantization results for different CFG scales with PixArt-$\alpha$.

We investigate FP4DiT's effectiveness under different CFG scales. Table 11 compares FP4DiT's quantization performance with PixArt-$\alpha$ for three different CFG scales, which are 3.0, 4.5 (default), and 7.0. Although FP4DiT is calibrated at CFG = 4.5, it remains robust and exhibits only minor degradation under different CFG scales, which is still outstanding among the baseline methods as shown in Table 2.

## A.3 Extended Experimental Settings

### A.3.1 Floating Point Quantization Scheme

Since Floating Point Quantization (FPQ) is not as straightforward as INT quantization, there has not been a simple and unified algorithm for performing FPQ yet. In this paper, we apply Algorithm 1 from (Kuzmin et al., 2022) to perform the FPQ. Unlike (Kuzmin et al., 2022) learning the clipping value $maxval$ and bit allocations between mantissa and exponent part, we use the absolute maximum of the tensor as $maxval$ and perform FPQ with a predetermined FP format.

In addition to the FPQ algorithm and the group-wise/token-wise quantization, we provide our quantization hyperparameters in Table 12. Note that the calibration size refers to the number of images we sampled from MS-COCO. For each image, we sample its input latent noise across 20 timesteps (50 for Hunyuan) as the calibration data for AdaRound. The activation FP format is for W4A6/W4A8 respectively.

## A.4 Evaluation Settings

| Model | Cali. Size | Cali. Step | Weight Format | Act. Format |
|---|---|---|---|---|
| PixArt-$\alpha$ | 128 | 2500 | E2M1 | E2M3/E3M4 |
| PixArt-$\Sigma$ | 128 | 2500 | E1M2 | E2M3/E3M4 |
| Hunyuan | 64 | 2500 | E2M1 | E2M3/E3M4 |

Table 12: Quantization calibration hyperparameters for FP4DiT.

For CLIP score (Hessel et al., 2021), we apply the openai-clip [1] library to measure the CLIP score between prompts and generated images. We employ ViT-B/32 as the CLIP model. To measure HPSv2 (Wu et al., 2023a) score, we generate 3.2k images across the four categories (800 images per category) and use the provided human preference predictor to estimate the performance. For the Fréchet Inception Distance (FID) (Heusel et al., 2017) measurement, we apply clean-fid [2] library to measure the FID.

## A.5 Hardware and Software Resources

We execute our FP4DiT and baseline experiments on two rack servers. The first is equipped with 2 Nvidia A100 80 GPUs, an AMD EPYC-Rome Processor and 512GB RAM. The second server has 8 Nvidia V100 32GB GPUs, an Intel Xeon Gold GPU and 756GB RAM.

Our code is running under Python 3 using Anaconda virtual environments and open-source repository forks based on Q-Diffusion (Li et al., 2023). We modified the code to implement our FP4DiT and enable the interface between Quantization scripts and the Hugging Face Diffusers [3] library. The hardware cost measurement is conducted using the ptflops [4] library. We provide a code implementation with README listing all necessary details and steps.

## A.6 Additional Visualization Results

In this section, we present the random samples derived from full-precision and W4A6/W4A8 PixArt-$\alpha$, PixArt-$\Sigma$, and Hunyuan that are quantized by baselines and FP4DiT. As depicted by Figures 11, 12, 13, 14, 15, and 16, FP4DiT generates results of impressive visual content. FP4DiT consistently shows superior performance across various DiT models. We list a detailed comparison between FP4DiT and the baselines:

1. PixArt-$\alpha$: On W4A8 (Fig. 11), FP4DiT generates more fine-grained images, e.g., macaroon and wave details. Besides, the baselines fail to generate a hedgehog while FP4DiT correctly depicts it. On W4A6 (Fig. 12), our method shows near W4A8 performance, unlike the noisier baselines.

2. PixArt-$\Sigma$: On W4A8 (Fig. 13), there is almost no noise on FP4DiT's generated images, which demonstrates the improvement of our method. As for the image details, FP4DiT has a more natural yoga posture, a more fine-grained model face, and more detailed cat hair and whiskers. Similar to PixArt-$\alpha$, FP4DiT's W4A6 (Fig. 14) images have the least noise and best image quality.

3. Hunyuan: On W4A8 (Fig. 15), FP4DiT's generated images closely align with the full precision model. For example, in the first image, FP4DiT has the same color right face while Q-Diffusion and TMFQ-DM alter the face's color. In the third image, FP4DiT generates the most similar face decoration. In addition, FP4DiT also has the least noise on Hunyuan DiT. On W4A6 (Fig. 16), all methods generate blur images. However, the contour of images identified from FP4DiT's visualization results matches the full precision images, while other baselines fail to produce a prompt-adherent contour.

---

[1] https://github.com/openai/CLIP
[2] https://github.com/GaParmar/clean-fid
[3] https://huggingface.co/docs/diffusers/en/index
[4] https://github.com/sovrasov/flops-counter.pytorch

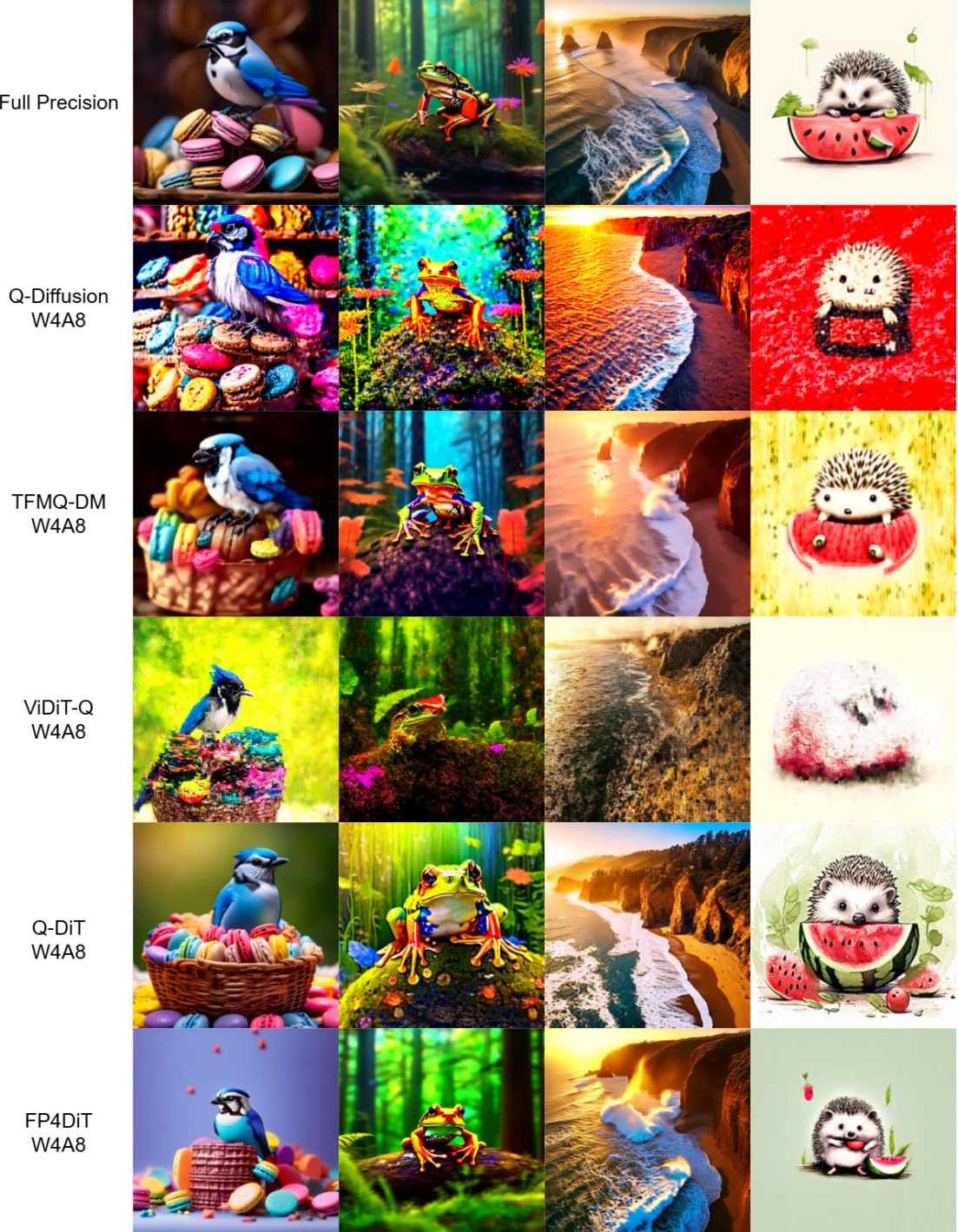

Figure 11: More visualization result for W4A8 PixArt-$\alpha$. Prompts: 'A blue jay standing on a large basket of rainbow macarons.'; 'Frog, in forest, colorful, no watermark, no signature, in forest, 8k.'; 'Drone view of waves crashing against the rugged cliffs along Big Sur's Garay Point beach. The crashing blue waters create white-tipped waves, while the golden light of the setting sun illuminates the rocky shore.'; 'An ink sketch style illustration of a small hedgehog holding a piece of watermelon with its tiny paws, taking little bites with its eyes closed in delight. Photo of a lychee-inspired spherical chair, with a bumpy white exterior and plush interior, set against a tropical wallpaper.'

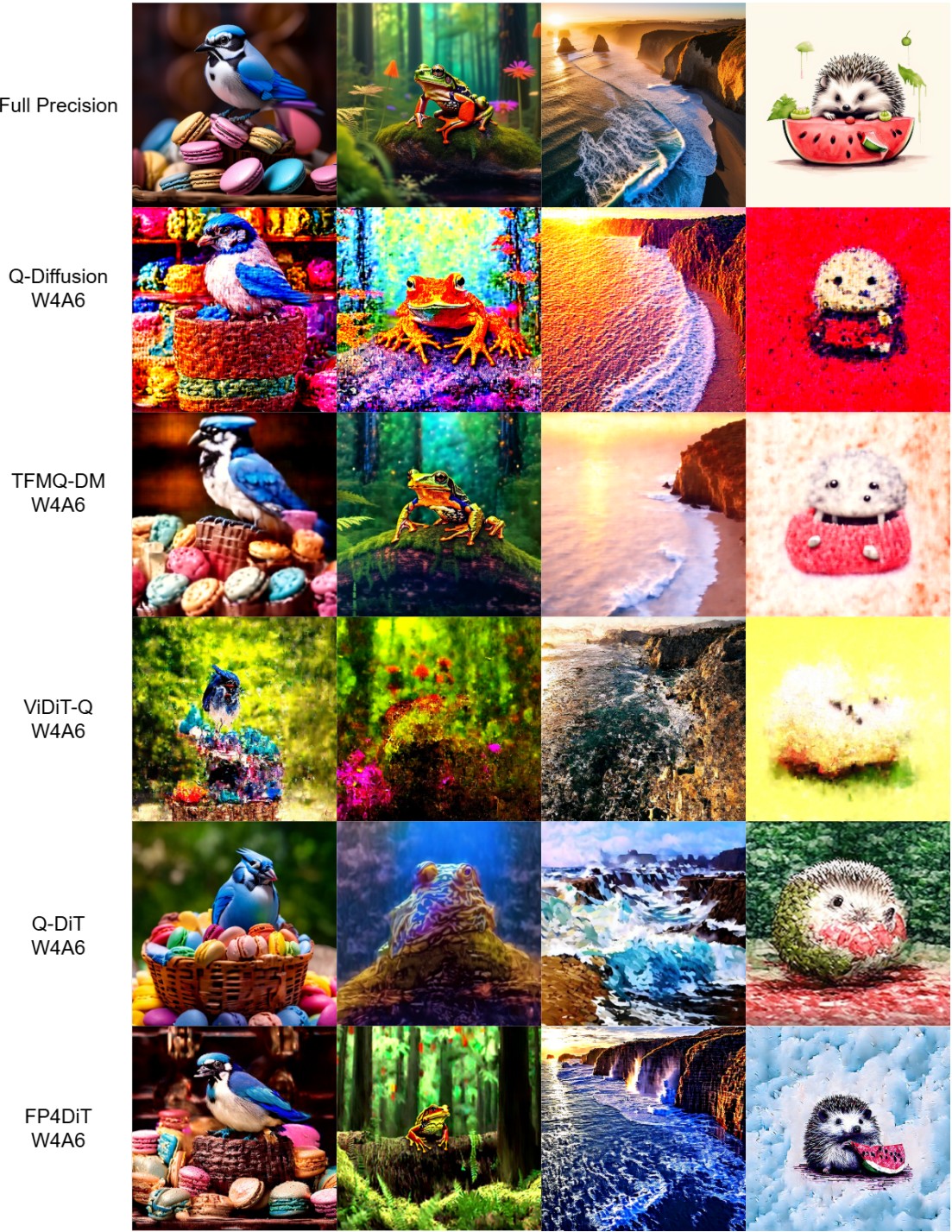

Figure 12: More visualization result for W4A6 PixArt-$\alpha$. Prompts: 'A blue jay standing on a large basket of rainbow macarons.'; 'Frog, in forest, colorful, no watermark, no signature, in forest, 8k.'; 'Drone view of waves crashing against the rugged cliffs along Big Sur's Garay Point beach. The crashing blue waters create white-tipped waves, while the golden light of the setting sun illuminates the rocky shore.'; 'An ink sketch style illustration of a small hedgehog holding a piece of watermelon with its tiny paws, taking little bites with its eyes closed in delight. Photo of a lychee-inspired spherical chair, with a bumpy white exterior and plush interior, set against a tropical wallpaper.'

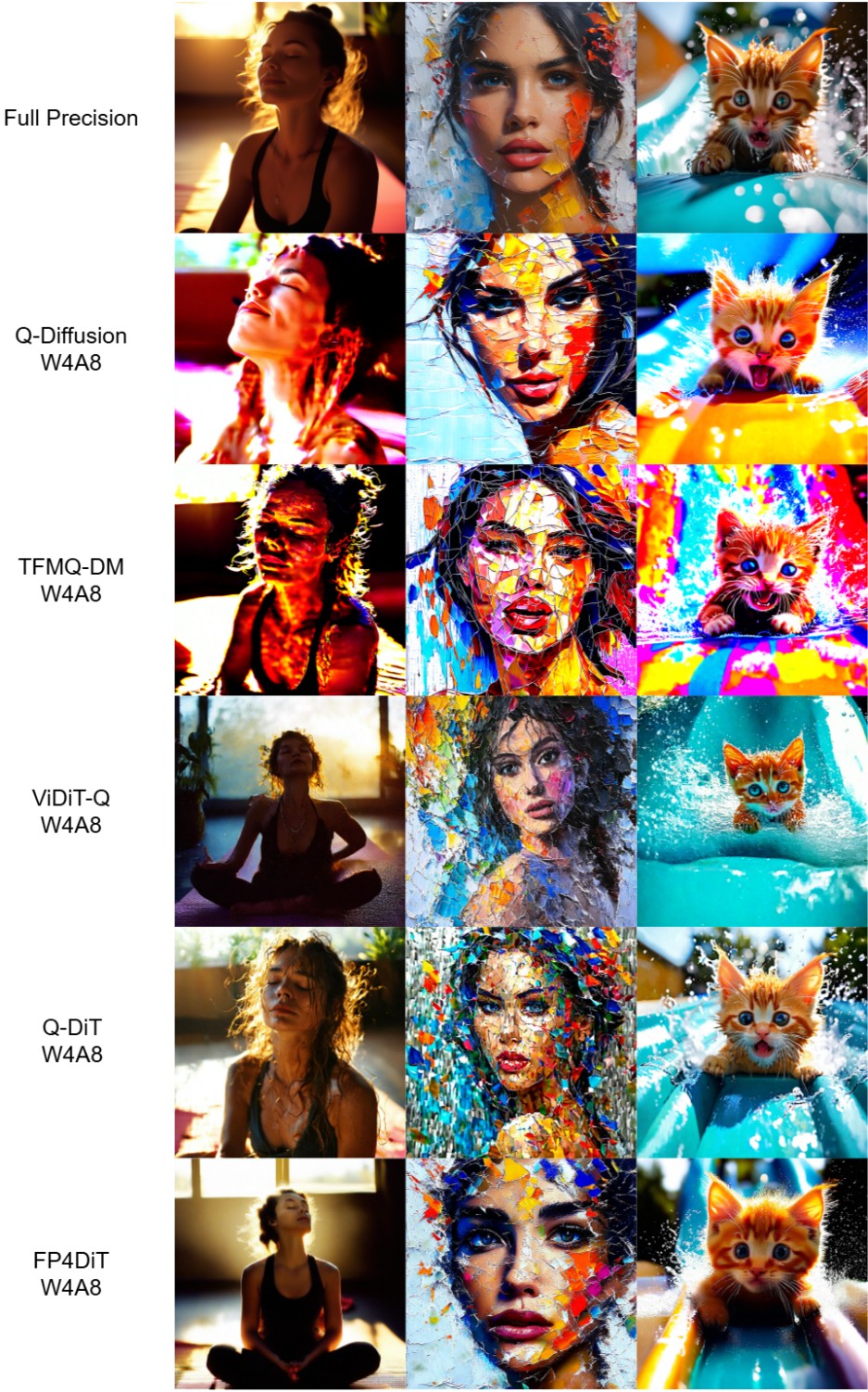

Figure 13: More visualization result for W4A8 PixArt-Σ. Prompts: 'A very attractive and natural woman, sitting on a yoka mat, breathing, eye closed, no make up, intense satisfaction, she looks like she is intensely relaxed, yoga class, sunrise, 35mm.'; 'Realistic oil painting of a stunning model merged in multicolor splash made of finely torn paper, eye contact, walking with class in a street.'; 'A cute orange kitten sliding down an aqua slide. happy excited. 16mm lens in front. we see his excitement and scared in the eye. vibrant colors. water splashing on the lens.'

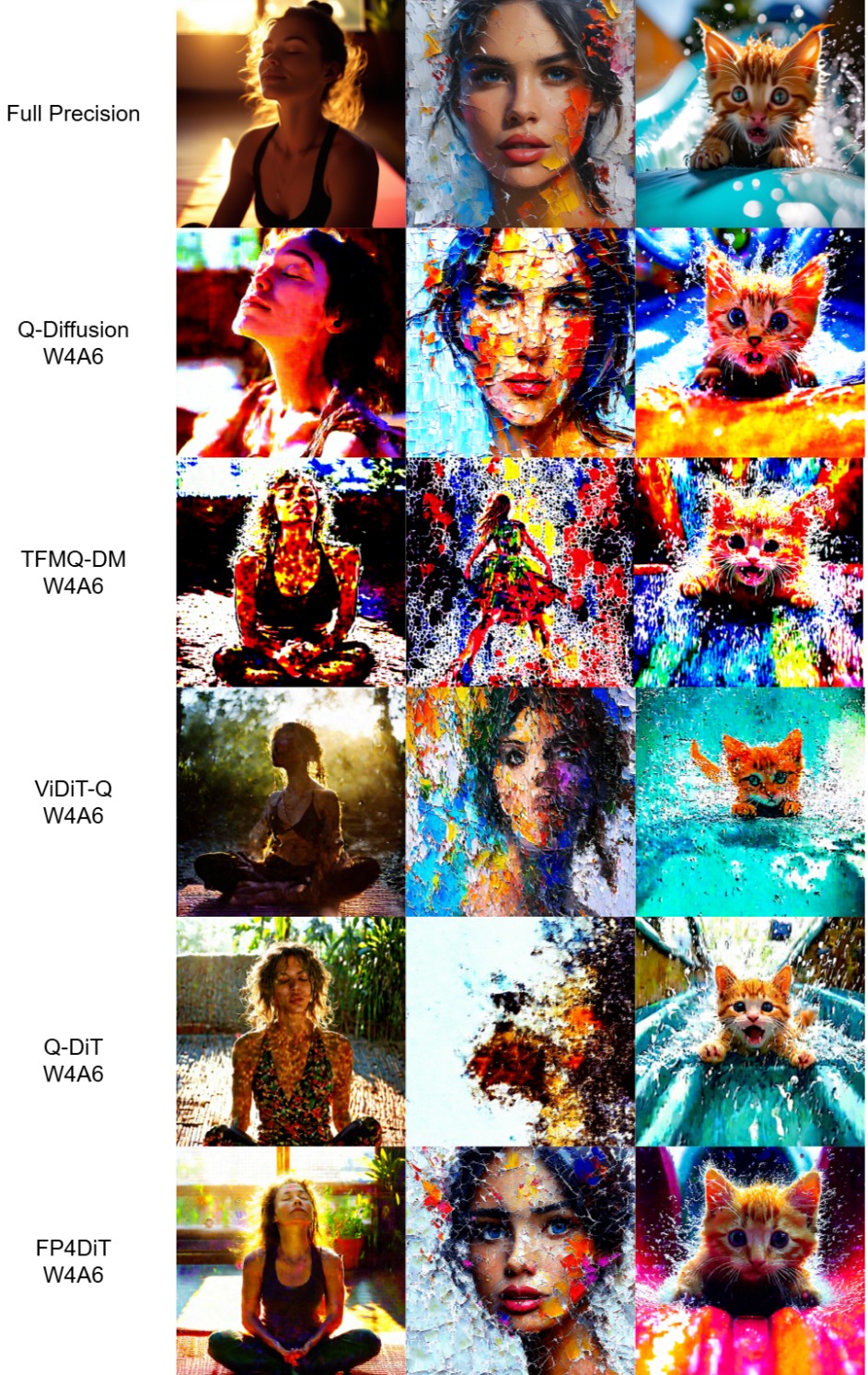

Figure 14: More visualization result for W4A6 PixArt-Σ. Prompts: 'A very attractive and natural woman, sitting on a yoka mat, breathing, eye closed, no make up, intense satisfaction, she looks like she is intensely relaxed, yoga class, sunrise, 35mm.'; 'Realistic oil painting of a stunning model merged in multicolor splash made of finely torn paper, eye contact, walking with class in a street.'; 'A cute orange kitten sliding down an aqua slide. happy excited. 16mm lens in front. we see his excitement and scared in the eye. vibrant colors. water splashing on the lens.'

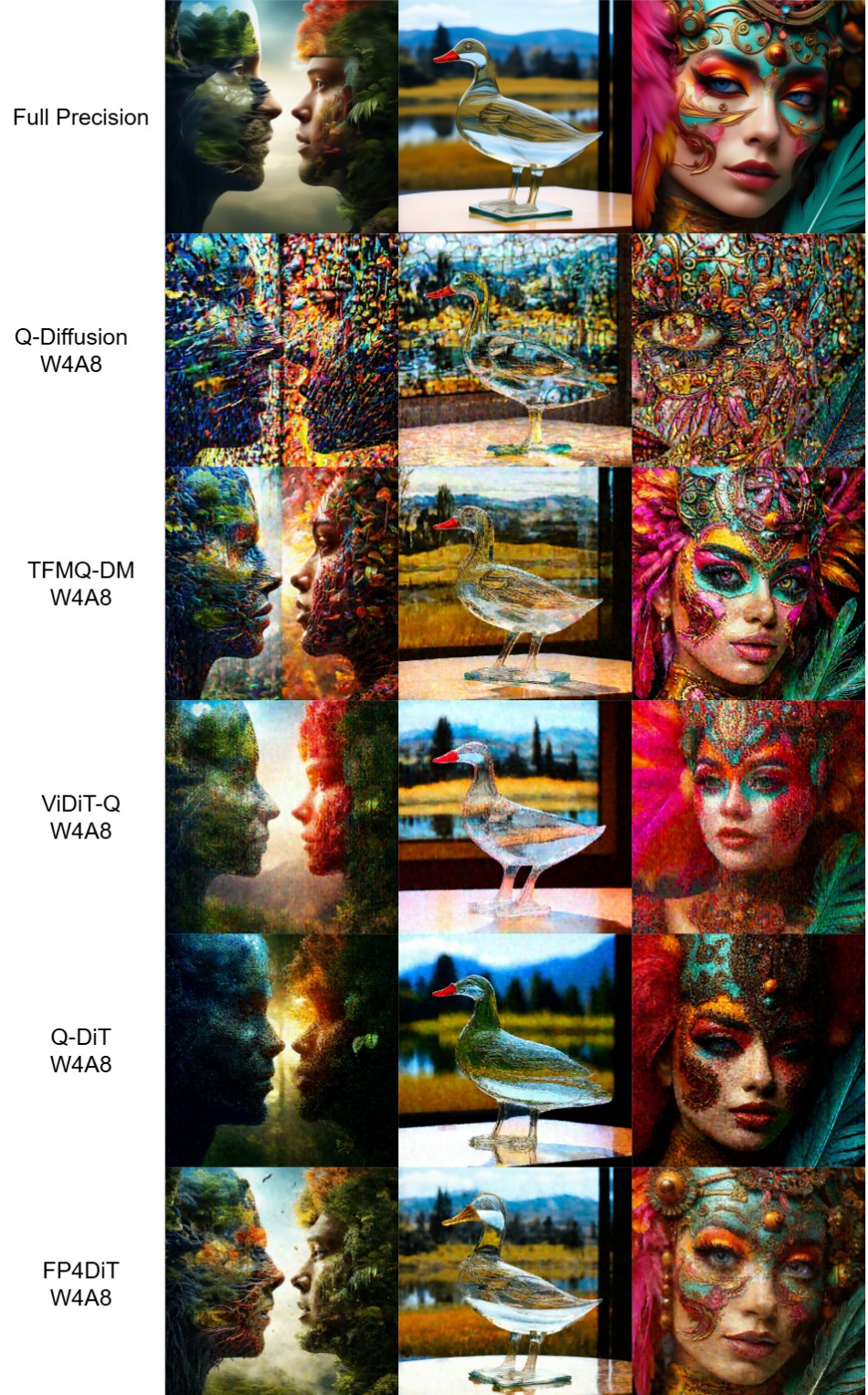

Figure 15: More visualization result for W4A8 Hunyuan. Prompts: 'nature vs human nature, surreal, UHD, 8k, hyper details, rich colors, photograph.'; 'A transparent sculpture of a duck made out of glass. The sculpture is in front of a painting of a landscape.'; 'Steampunk makeup, in the style of vray tracing, colorful impasto, uhd image, indonesian art, fine feather details with bright red and yellow and green and pink and orange colours, intricate patterns and details, dark cyan and amber makeup. Rich colourful plumes. Victorian style.'

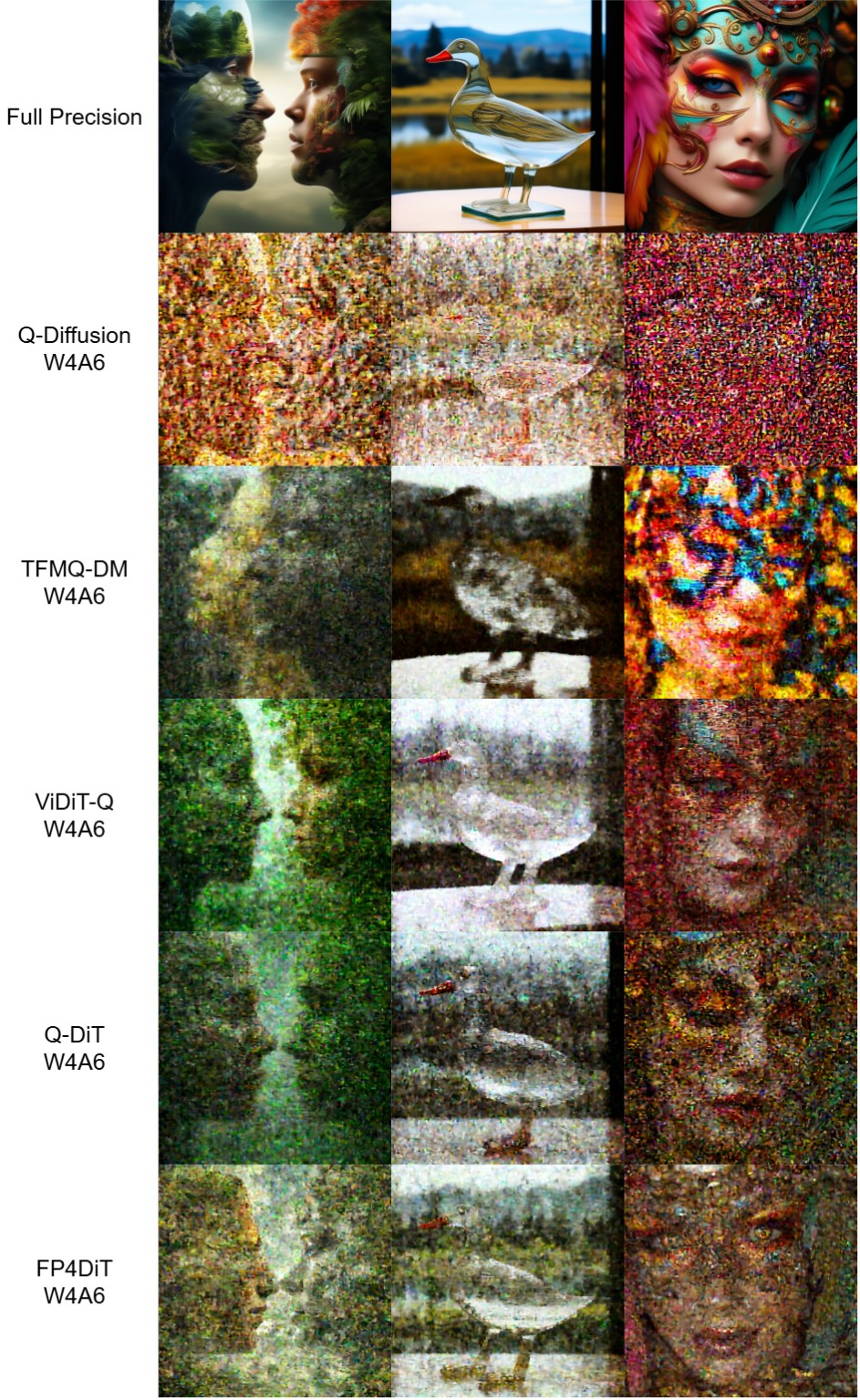

Figure 16: More visualization result for W4A6 Hunyuan. Prompts: 'nature vs human nature, surreal, UHD, 8k, hyper details, rich colors, photograph.'; 'A transparent sculpture of a duck made out of glass. The sculpture is in front of a painting of a landscape.'; 'Steampunk makeup, in the style of vray tracing, colorful impasto, uhd image, indonesian art, fine feather details with bright red and yellow and green and pink and orange colours, intricate patterns and details, dark cyan and amber makeup. Rich colourful plumes. Victorian style.'

