# OpenReview forum: "FP4DiT: Towards Effective Floating Point Quantization for Diffusion Transformers"
_TMLR — Accepted by TMLR_

### Review · Reviewer_JPJW · 2025-07-24

**Summary Of Contributions:**

This paper proposes a method to quantize diffusion transformers, covering both weight and activation quantization. For weighs, the authors propose a scale-aware adaptive rounding method to achieve a 4-bit floating-point quantization applied heterogeneously throughout the denoising networking. For activations, the authors consider both 8-bit and 6-bit floating-point formats with per token scaling. Evaluation is conducted on two PixArt models and Hunyuan, using HPSv2, MS-COCO with FID and CLIP, and human evaluation. Overall, the method achieves a good speedup of 1.5x while improving generative image quality over prior methods.

Strengths:
- The methodology is well motivated by compelling evaluation of the DiT’s time-step dependent activation behavior, and shortcomings of prior uniform quantization methods.
- The idea of using E3M0 for the input projection before the GELU activations is interesting, and well-reasoned.
- The authors perform comprehensive evaluations, covering multiple models and metrics.

Weaknesses:
- While surpassing prior methods in many cases, the image quality is still lacking with visible quantization artifacts in W4A6 and in some W4A8 cases.
- Several of the claims are overstated, such as superior performance, where the metric improvements may not be statistically significant.
- Some of the experimental setups are not well described, requiring inspection of source code for reproduction.

**Additional Comments:**

Questions:
- The motivation for E3M0 prior to the GELU is to capture the negative output region near zero. Would this still be necessary if the GELU FFNs were hardened with ReLU as is sometimes done with vision transformers for edge devices? And would models that utilize SiLU prefer a different quantization?
- Given the degradation of W4A6 vs W4A8, and the lack of existing hardware support, is there a reason to prefer the 6-bit activations over 8-bit? It seems to me that the tradeoff is not worthwhile.

Suggestions:
- There should be an explanation (including the abstract) of what W4A8 means; it may be unclear to readers unfamiliar with the notation.
- Replacing the Northern Lights image example in Figure 7 may make the quantization results easier to observe – Landscapes are not ideal for visual evaluation, as they are relatively easy for even small models to generate convincingly.
- Figure 4 and Figure 5 should have subfigure labels to easily see what each represents without reading the full caption.

**Audience:**

Yes

**Audience Explanation:**

This paper explores several behavior characteristics of diffusion transformers and their impact on weight quantization. The novel method of scale-aware quantization, and the improved results at W4A6 (despite the image quality loss) are also interesting and will be helpful to future quantization works.

**Broader Impact Concerns:**

No concerns.

**Claims And Evidence:**

No

**Claims Explanation:**

Many of the quality claims have over-emphasized significance, where they instead demonstrate matching or slightly improved behavior compared to baselines. For example, it is not clear that an overall HPSv2 score of 27.78 vs 27.52 is statistically significant.
Additionally, the evaluation methodology on FID and CLIP deviates from standard practices. FID on MS-COCO is typically done with 30k samples to improve the reliability and stability of the score. CLIP typically uses VIT-B/16 on 30k samples as proposed by GLIDE. Notably, the authors failed to reproduce the FID result of PixArt-$\alpha$ (34.05) as reported in the original paper (7.32).

**Requested Changes:**

For acceptance:
- Improve the accuracy of the FID and CLIP metrics with more samples. The full precision approaches should reproduce (or come close to) the previously reported metrics or explain why a deviation occurs.
- Soften the claims of superior performance. The FID and CLIP results are better, while the HPSv2 results are marginal.
- Reduce the computational improvement claim or better position the motivation in the introduction and related works. Currently, it reads as quantization being the only viable approach, yet FP4DiT only achieves a 1.5x speedup. Timestep distillation and model scale distillation / pruning are orthogonal methods that can have greater speedups on edge devices.
- Improve the experimental setup information. What precision are the MACs done in? What precision do the group and token scale coefficients use? How were the results in Table 6 computed (e.g., sample size, batch size, resolution, etc.)?

To Strengthen:
- Understanding why the speedup is only 1.5x for W4A8 would be helpful. Is it truly computationally bound since the memory movement is reduced by 4x for the weights and 2x for the activations? Which layer is the bottleneck?
- When looking at the CLIP impact vs calibration steps, the curve appears to be increasing from 10k to 20k. Could 40k potentially outperform 2.5k samples?
- Is there a dependency on the calibration set used? MS-COCO is lower-resolution compared to PixArt-$\Sigma$ and Hunyuan and lacks the variability of the pre-training datasets.
- Investigation into FP4DiT’s performance on timestep distilled models would also be appreciated (e.g., PixArt-$\delta$).
- Understanding why Hunyuan experienced such a large FID and CLIP degradation when moving to W4A6 could be impactful, since the same extreme was not observed in the PixArt models.

---

### Review · Reviewer_cbZf · 2025-07-26

**Summary Of Contributions:**

This work focuses on the post-training quantization (PTQ) of text-to-image (T2I) diffusion models (DMs) and proposes a floating-point quantization framework, FP4DiT. Considering the non-uniform distribution of weights and activations in DMs, floating-point quantization can reduce the quantization error compared to the existing INT-based framework. The author also fully investigates the GELU activation function and locates that sensitive intervals are important. The proposed FP4DiT is built upon AdaRound, and the author improves it by introducing a scaling tailored for FP quantization and an effective online activation quantization scheme. The proposed FP4DiT is verified across two benchmarks (HPSv2 and MS-COCO) and various DMs (PixArt-$\alpha$, $\Sigma$, Hunyuan), demonstrating its effectiveness and efficiency.

**Audience:**

Yes

**Audience Explanation:**

The topic of Diffusion Model quantization is important and under-investigated compared to traditional small-scale models (CNNs, ViTs) and LLMs. I believe TMLR's audience working on the efficiency of GenAI will definitely be interested in this paper.

**Claims And Evidence:**

Yes

**Claims Explanation:**

This paper conducts a full investigation and reveals some important challenges and bottlenecks of DM PTQ. Both empirical results and theoretical analysis show that FP4DiT would be a solution to the challenges. The theoretical proof of Scale-Aware AdaRound (in the appendix) and hardware cost comparison are particularly impressive. The design of ablation experiments is reasonable, showing how the proposed components of FP4DiT improve the naive FPQ of DMs. In addition, the presentation quality of this work is good, whole paper is easy to follow with clear figures and tables.

**Requested Changes:**

- Although the proposed FP4DiT is designed for DiT-based DMs, it seems that it could also be directly applied to UNet-based DMs without much modification. The results on UNet-based DMs, especially SD series (SDXL, SD3), would be interesting and helpful as they are more widely adopted in real applications.
- While cfg changes, what will be the performance of FP4DiT? Please include related discussions and experiments since this is an important setting in DMs.
- More recent and widely-used benchmarks and metrics could be included in experiments, such as GenEval [1], DPG-bench [2], and Image Reward [3].

[1] GenEval: An Object-Focused Framework for Evaluating Text-to-Image Alignment

[2] ELLA: Equip Diffusion Models with LLM for Enhanced Semantic Alignment

[3] ImageReward: Learning and Evaluating Human Preferences for Text-to-image Generation, NeurIPS 2023

---

### Review · Reviewer_nNHa · 2025-09-11

**Summary Of Contributions:**

FP4DiT adapts PTQ to Diffusion Transformers with per-layer mixed FP4 formats, a scale-aware AdaRound, and token-wise activation quantization, yielding near-lossless W4A8, competitive W4A6 on three DiTs, and ~1.5× layer-level speedups with lower memory on Blackwell GPUs.

Strengths:
1. FP4DiT adapts FP formats to DiT by introducing mixed FP4 formats (E3M0/E2M1/E1M2 per layer), which better capture the sensitive range of GELU activations.
2. FP4DiT presents an enhanced rounding and scaling strategy designed for FP quantization.
3. Experiments span three representative DiT models with standard T2I benchmarks (e.g., FID, CLIP, HPSv2), include ablations on calibration budget and format choice, and report real-hardware measurements on Blackwell GPUs showing ~1.5× speedups.

Weaknesses:
1. The novelty is limited. The contribution primarily adapts established FPQ to DiT; the main new component is scale-aware AdaRound.
2. Human eval is small-scale: Preference studies (15–17 participants) lack statistical significance tests, inter-rater agreement, and detailed blinding/randomization protocols。
3. While W4A8 is near-lossless, W4A6 shows larger drops on challenging setups.

**Audience:**

Yes

**Audience Explanation:**

Given the scarcity of prior work on applying FPQ to DiT architectures, this paper fills a timely gap and should serve as a useful reference for subsequent efforts in DiT quantization.

**Claims And Evidence:**

Yes

**Claims Explanation:**

1. On three DiT models, FID/CLIP/HPSv2 say W4A8 is basically lossless.
2. The improvements mainly come from scale-aware AdaRound with mixed FP4, as confirmed by the calibration/format ablations

**Requested Changes:**

1. More ablation study about block-wise optimization will enhance this paper.

2. End-to-end latency analysis will better than per-layer latency analysis.

3. Additional DiT families and harder settings (higher resolutions, longer/complex or multilingual prompts) will demonstrate generality. (not necessary)

4. Deepen the quantization discussion will be better. This paper lacks many quantization references, such as OBQ [1], COMQ [2], QuIP [3], SpinQuant [4], MagR [5].

[1] Frantar, Elias, and Dan Alistarh. "Optimal brain compression: A framework for accurate post-training quantization and pruning." Advances in Neural Information Processing Systems 35 (2022): 4475-4488.

[2] Zhang, Aozhong, et al. "Comq: A backpropagation-free algorithm for post-training quantization." IEEE Access (2025).

[3] Chee, Jerry, et al. "Quip: 2-bit quantization of large language models with guarantees." Advances in Neural Information Processing Systems 36 (2023): 4396-4429.

[4] Liu, Zechun, et al. "Spinquant: Llm quantization with learned rotations." arXiv preprint arXiv:2405.16406 (2024).

[5] Zhang, Aozhong, et al. "Magr: Weight magnitude reduction for enhancing post-training quantization." Advances in neural information processing systems 37 (2024): 85109-85130.

---

> ### Author Response · Authors · 2025-09-25
>
> **Weaknesses**
>
> > **The novelty is limited. The contribution primarily adapts established FPQ to DiT; the main new component is scale-aware AdaRound.**
>
> We thank the reviewer for the assessment of our work, especially claiming our work will be interesting to the TMLR’s audience. Notably, FP4DiT is the first to apply FPQ to DiT and successfully and jointly optimized weight calibration technique (AdaRound), the FP format selection and online activation to achieve the best performance under W4A8 and W4A6 in the DiT quantization literature, with a bunch of insights provided.  While we do not claim our method represents a fundamental breakthrough in quantization techniques, we believe this work meets the TMLR’s evaluation criteria by offering a technically outstanding contribution to DiT’s lightweighted deployment and quantization.
>
> For reference only, the following is excerpted from TMLR acceptance criteria:
> "Nor should it form the basis for rejecting work on a method considered not “novel enough”, as novelty of the studied method is not a necessary criteria for acceptance. We explicitly avoid these terms (“significant”, “impactful”, “novel”), and focus instead on the notion of “interest”. If the authors make it clear that there is something to be learned by some researchers in their area from their work, then the criterion of interest is considered satisfied.”
>
> > **Human eval is small-scale: Preference studies (15–17 participants) lack statistical significance tests, inter-rater agreement, and detailed blinding/randomization protocols.**
>
> 1. First, note that existing DiT quantization methods [1, 2, 3] only rely on quantitative metrics for evaluation. To the best of our knowledge, BitsFusion [4] done by Snap Inc. is the only exception in the literature that conducted a human preference study for diffusion model quantization (but for UNet only, is closed sourced, and is not for DiT). BitsFusion conducted 1,632 pairwise preference comparisons in their evaluation, which aligns with the scale of our preference study (i.e., we conducted 2040 and 1125 1-of-3 preference comparisons in our preference studies for Tables 3 and 4, respectively).
>
> 2. Second, our human preference study already takes into account the inter-rater agreement, as each prompt is evaluated by multiple participants (voters) and their aggregated preferences are reported in Tables 3 and 4. For rigor and fairness, we also adopted a blind-review protocol, since voters were not informed which image was generated by which method in evaluation.
>
> 3. Lastly, as our goal is to show which method (our method or baseline methods) is voted more by human evaluators (under blind review), we use preference study (by multiple voters) instead of significance test which is more suitable to test some statistical hypothesis. Similarly, BitsFusion did not do significance tests either which is not applicable here. The preference voting outcomes are consistent with our quantitative metrics in Table 2, where our method attains the strongest performance on all reported metrics. Furthermore, the superior visual result examples provided in Figures 7, 8, 11, 12, 13, 14, 15, and 16 also match closely the outcomes of our human preference study; there is visible quality gap between our method and the baselines, confirming voters’ preferences.
>
> > **While W4A8 is near-lossless, W4A6 shows larger drops on challenging setups.**
>
> 1. W4A6 is more challenging because the aggressive reduction in activation precision increases quantization difficulty. As a result, all quantization methods exhibit a performance drop when transitioning from W4A8 to W4A6.
>
> 2. Despite this inherent challenge, our method consistently outperforms all baseline techniques on the more challenging W4A6 setting. In particular, while other baselines exhibit severe degradation that leads to broken or unusable images, our method maintains acceptable image quality and achieves the best performance among all compared approaches. As shown in Table 2, under W4A6, FP4DiT attains the highest scores across all quantitative metrics on PixArt DiT, and in Figure 7, it produces the most visually coherent images with the closest resemblance to the full-attention results.
>
> 3. As for the Hunyuan DiT,  A6 quantization remains an open challenge, as none of the methods can achieve satisfactory performance.
>
> [1] Li et al. "Q-diffusion: Quantizing diffusion models." Proceedings of the IEEE/CVF International Conference on Computer Vision. 2023.
>
> [2] Huang et al. "Tfmq-dm: Temporal feature maintenance quantization for diffusion models." Proceedings of the IEEE/CVF Conference on Computer Vision and Pattern Recognition. 2024.
>
> [3] Zhao et al. "Vidit-q: Efficient and accurate quantization of diffusion transformers for image and video generation." arXiv preprint arXiv:2406.02540 (2024).
>
> [4] Sui et al. "Bitsfusion: 1.99 bits weight quantization of diffusion model." Advances in Neural Information Processing Systems 37 (2024): 76775-76818.

---

> ### Author Response · Authors · 2025-09-25
>
> **Requested Changes**
>
> >**More ablation study about block-wise optimization will enhance this paper.**
>
> We add an ablation study about the effect of calibration dataset size N on PixArt-$\alpha$ to the Appendix. The result shows the HPSv2 score of the W4A8 PixArt-$\alpha$ quantized by our method at different calibration dataset sizes N. The W4A8 model quantized by our method shows steady gains up to $N=256$, with scores plateauing thereafter. This indicates that a relatively small calibration set is sufficient, and we adopt $N=128$ as the default to balance efficiency and performance.
>
> In addition to that, we also conduct the ablation studies about the quantization group size and classifier-free guidance (CFG) in the Appendix. Specifically, we vary the group size to analyze its impact on accuracy–efficiency trade-offs, showing how finer grouping improves fidelity at the cost of higher computation, while coarser grouping yields better compression with minor quality degradation. For CFG, we examine different guidance scales to evaluate their influence on sample diversity and perceptual quality, demonstrating that our quantized model maintains stable generation behavior across a wide range of CFG values.
>
> >**End-to-end latency analysis will better than per-layer latency analysis.**
>
> FPQ is a more advanced quantization technique than standard integer quantization. Thus, low-bit floating-point support is only emerging, making end-to-end kernel development currently impractical. Therefore, building a complete end-to-end FPQ kernel stack is beyond the scope of this work. The focus of this paper is the fundamental FPQ method and its efficiency. In this context, our per-layer (kernel-level) latency analysis is an appropriate and sufficient measure to validate the effectiveness of FPQ.
>
> >**Additional DiT families and harder settings (higher resolutions, longer/complex or multilingual prompts) will demonstrate generality. (not necessary)**
>
> 1. Our experiments already cover both simple and complex prompts to ensure a balanced evaluation. Specifically, we use MS-COCO prompts which are short descriptive sentences (e.g. A man riding a skateboard down a city street), following established literature [1, 2] in diffusion model benchmarking. We further evaluate our method on HPSv2 prompts [3], which contain real prompts that are written by the Stable Diffusion users [4]. They are longer and more descriptive compared to MS-COCO prompts (e.g. Lionel Messi portrayed as a sitcom character). Finally, we include user-provided prompts from PixArt, which are the most complex and diverse (multi-sentence and stylistically detailed), to provide qualitative visualization comparison (see Figures 7, 8, 11, 12, 13, 14, 15, 16). Across all three prompt sets, our method consistently outperforms the baselines.
>
> 2. While other DiT quantization literature [1, 2, 5, 6] generates 256x256 and 512x512 resolution images, in our experiment, we use 512x512 and 1024x1024 as the image resolution, thereby our method addresses more challenging and practical resolution settings.
>
> >**Deepen the quantization discussion will be better. This paper lacks many quantization references, such as OBQ, COMQ, QuIP, SpinQuant, MagR**
>
> We have revised the manuscript to include these quantization references. However, as these methods are designed specifically for LLM quantization and they cannot be applied to DiT models. Instead, we comprehensively evaluate our approach against the state-of-the-art DiT quantization methods and demonstrate consistent improvements over them.
>
> [1] Li et al. "Q-diffusion: Quantizing diffusion models." Proceedings of the IEEE/CVF International Conference on Computer Vision. 2023.
>
> [2] Huang et al. "Tfmq-dm: Temporal feature maintenance quantization for diffusion models." Proceedings of the IEEE/CVF Conference on Computer Vision and Pattern Recognition. 2024.
>
> [3] Wu, Xiaoshi, et al. "Human preference score v2: A solid benchmark for evaluating human preferences of text-to-image synthesis." arXiv preprint arXiv:2306.09341 (2023).
>
> [4] Wang, Zijie J., et al. "Diffusiondb: A large-scale prompt gallery dataset for text-to-image generative models." arXiv preprint arXiv:2210.14896 (2022).
>
> [5] Zhao et al. "Vidit-q: Efficient and accurate quantization of diffusion transformers for image and video generation." arXiv preprint arXiv:2406.02540 (2024).
>
> [6] Chen, Lei, et al. "Q-dit: Accurate post-training quantization for diffusion transformers." Proceedings of the Computer Vision and Pattern Recognition Conference. 2025.

---

### Decision · Action_Editor_GrKn · 2025-10-15

**Recommendation:** Accept as is

**Additional Comments:**

This paper investigates post-training quantization of diffusion transformer models (DiT) using low-precision floating-point formats. By analyzing the distributions of weights and activations, the paper proposes techniques such as mixed-precision settings and adaptive rounding methods to quantize DiT models to W4A6 and W4A8 precision. The proposed methods are evaluated on popular DiT models and benchmarks, demonstrating improved performance.

The paper is reasonably well written and addresses an important practical problem. The evaluation is fairly comprehensive. However, although performance improvements are shown, the gains are not substantial compared to prior work, and the proposed methods are not particularly novel.

Nevertheless, the paper provides useful insights that can benefit researchers working on DiT model acceleration, including:

1.	Data distribution analysis

2.	Mixed-precision configurations

3.	Extension of adaptive rounding to logarithmic/float domains

4.	Evaluation on various DiT models and benchmarks

Therefore, the paper is recommended for acceptance.

**Audience:**

Yes

**Audience Explanation:**

This paper provides empirical insights and practical techniques to improve low-precision quantization of diffusion transformer (DiT) models, which can benefit researchers working on DiT model acceleration.

**Claims And Evidence:**

Yes

**Claims Explanation:**

The claims are reasonably well supported.